# Independent insulin signaling modulators govern hot avoidance under different feeding states

**Meng-Hsuan Chiang**[1�], **Yu-Chun Lin**[1,2☹], **Sheng-Fu Chen**[3], **Peng-Shiuan Lee**[2], **Tsai-Feng Fu**[4], **Tony Wu**[5], **Chia-Lin Wu**[1,5,6,7]*

**1** Graduate Institute of Biomedical Sciences, College of Medicine, Chang Gung University, Taoyuan, Taiwan, **2** Department of Biomedical Sciences, College of Medicine, Chang Gung University, Taoyuan, Taiwan, **3** NHRI Institute of Biomedical Engineering & Nanomedicine, Miaoli, Taiwan, **4** Department of Applied Chemistry, National Chi Nan University, Nantou, Taiwan, **5** Department of Neurology, New Taipei Municipal TuCheng Hospital, Chang Gung Memorial Hospital, New Taipei City, Taiwan, **6** Department of Biochemistry, College of Medicine, Chang Gung University, Taoyuan, Taiwan, **7** Brain Research Center, National Tsing Hua University, Hsinchu, Taiwan

☹ These authors contributed equally to this work.
* clwu@mail.cgu.edu.tw

**Data Availability Statement:** All relevant data are within the paper and its Supporting Information files.

## Abstract

Thermosensation is critical for the survival of animals. However, mechanisms through which nutritional status modulates thermosensation remain unclear. Herein, we showed that hungry *Drosophila* exhibit a strong hot avoidance behavior (HAB) compared to food-sated flies. We identified that hot stimulus increases the activity of α′β′ mushroom body neurons (MBns), with weak activity in the sated state and strong activity in the hungry state. Furthermore, we showed that α′β′ MBn receives the same level of hot input from the mALT projection neurons via cholinergic transmission in sated and hungry states. Differences in α′β′ MBn activity between food-sated and hungry flies following heat stimuli are regulated by distinct *Drosophila* insulin-like peptides (Dilps). Dilp2 is secreted by insulin-producing cells (IPCs) and regulates HAB during satiety, whereas Dilp6 is secreted by the fat body and regulates HAB during the hungry state. We observed that Dilp2 induces PI3K/AKT signaling, whereas Dilp6 induces Ras/ERK signaling in α′β′ MBn to regulate HAB in different feeding conditions. Finally, we showed that the 2 α′β′-related MB output neurons (MBONs), MBON-α′3 and MBON-β′1, are necessary for the output of integrated hot avoidance information from α′β′ MBn. Our results demonstrate the presence of dual insulin modulation pathways in α′β′ MBn, which are important for suitable behavioral responses in *Drosophila* during thermoregulation under different feeding states.

## Introduction

Temperature directly affects several biological processes, such as enzymatic reactions within the body and reproduction. The avoidance of unfavorable environmental temperatures is an innate behavior in all animals, from tiny flatworms to largest fish in the world, the whale

**Funding:** This work was supported by grants from the National Science and Technology Council (112-2311-B-182-002-MY3 and 109-2326-B-182-001-MY3) to C-LW, Chang Gung Memorial Hospital (CMRPD1M0301-3, CMRPD1M0761-3, and BMRPC75) to C-LW. The funders had no role in the study design, data collection and analysis, decision to publish, or manuscript preparation.

**Competing interests:** The authors have declared that no competing interests exist.

**Abbreviations:** AC, anterior cell; AHL, adult hemolymph-like; Dilps, *Drosophila* insulin-like peptide; HAB, hot avoidance behavior; HC, hot cell; IPC, insulin-producing cell; lALT, lateral antennal lobe tract; LH, lateral horn; mALT, medial antennal lobe tract; MBn, mushroom body neuron; MBON, mushroom body output neuron; PAM, protocerebral anterior medial; PI, pars intercerebralis; PLP, posterior lateral protocerebrum; PPL, protocerebral posterior lateral; PWM, pulse-width modulation; ROI, region of interest; SEM, standard error of mean; sNPF, short neuropeptide F; t3ALT, transverse 3 antennal lobe tract.

sharks [1,2]. Thermosensation and temperature avoidance behavior are important for avoiding extremely hot or cold conditions and regulating the body temperature, both of which are critical for survival [3]. The fruit fly *Drosophila melanogaster* has a small size and is highly sensitive to external temperatures. Fruit flies prefer an ambient temperature (approximately 25°C) and are able to accurately detect a suitable environment for dwelling. Thermosensation in *Drosophila* relies on multiple classes of thermoreceptors in the last antennal segment of the arista [4–6]. There are at least 4 classes of thermoreceptors in *Drosophila*, including receptors for innocuous (harmless) heat, receptors for noxious (harmful) heat, receptors for innocuous cold, and receptors for noxious cold [4–6]. The hot cell (HC) neurons in the arista of the antenna [6] and the anterior cell (AC) neurons inside the fly brain [5] are majorly function as sensors for harmless hot stimulus. The hot stimulus is conveyed from the antenna lobe to the higher brain center, including the dendritic region of the mushroom bodies (MBs) called the calyx, lateral horn (LH), and posterior lateral protocerebrum (PLP) via the medial antennal lobe tract (mALT) [7]. In addition to mALT, the lateral antennal lobe tract (lALT) and transverse 3 antennal lobe tract (t3ALT) can also convey hot stimuli from the antenna lobe to the LH and PLP, but not to the MB [7].

Hunger is an uneasy or painful sensation due to food deprivation. The state of hunger triggers various animal behaviors to meet energy and nutritional requirements. Hunger reduces the core body temperature in mammals [8]. Insulin is a peptide hormone secreted by the pancreatic β-cells and plays a major role in energy homeostasis by regulating the blood glucose levels in humans [9]. In invertebrates, the evolutionary conserved insulin-like peptides play crucial roles in regulating metabolism, growth, and longevity. Approximately 40 insulin-like peptides have been identified in *Caenorhabditis elegans* [10]. In *D. melanogaster*, 8 insulin-like peptides (Dilps) and 1 Dilp receptor (InR) have been identified [11]. Different Dilps are produced by distinct cell types or tissues during different developmental and adult stages [11]. In the adult fly brain, 14 insulin-producing cells (IPCs) are located in the *pars intercerebralis* (PI) and express Dilp2, Dilp3, and Dilp5, whereas Dilp1 is additionally expressed in larval IPCs [12–15]. Dilp2 secretion is dependent on the nutritional status; moreover, nutrient deprivation inhibits the secretion of Dilp2 by IPCs [16,17]. Dilp6 is produced by the adult fat body, and *dilp6* mRNA levels are increased during starvation. The secretion of Dilp6 from the fat body is responsible for the starvation-induced reduction in preferred temperature (T*p*) [18]. In addition, overexpression of *dilp6* in the fat body represses the expression of *dilp2* and *dilp5* mRNA in the brain and reduces the secretion of Dilp2 [19].

MB is a huge brain structure comprising approximately 2,000 MB neurons (MBns), called Kenyon cells in each brain hemisphere, and can be further classified into αβ, γ, and α′β′ MBn according to the distribution of their axons [20]. Studies have demonstrated that MBn plays a role in *Drosophila* temperature preference behaviors via dopamine and cAMP signaling [21–23]. It has also been shown that MBn integrates satiety and hunger signals to regulate food-seeking behaviors [24,25]. However, whether satiety and hunger signals affect thermosensation and temperature preferences in flies, is still unclear. Herein, we showed that hungry flies exhibit a stronger hot avoidance behavior (HAB) compared to food-sated flies. However, hungry and sated flies exhibited no difference in cold avoidance behavior. We showed that hot signals are conveyed by the cholinergic mALT projection neurons, the axons of which are functionally connected to the dendritic region of α′β′ MBn. We also revealed that hot stimulus evokes the same level of calcium response in mALT projection neurons in sated and hungry flies. Our behavioral data suggest that MB activity is required for HAB and α′β′ MBn plays a crucial role in both sated and hungry states. Live brain imaging showed a stronger calcium response to hot stimuli in α′β′ MBn, particularly in the hungry state. Genetic expression of the constitutively active form of InR in α′β′ MBn inhibited the calcium response to hot stimuli

and consequently, reduces HAB. In addition, expressing the dominant-negative form of InR in α′β′ MBn caused the opposite effect, suggesting that insulin signaling in α′β′ MBn negatively regulates hot sensation. It has been shown that *dilp2* transcript represents approximately 80% of all *dilp* transcripts present in IPCs [26]. RNAi-mediated silencing of *dilp2* in IPCs increased HAB and α′β′ MBn activity only in the sated state. Interestingly, *dilp6* silencing in the fat body increased HAB and α′β′ MBn activity, specifically in the hungry state. We further showed that PI3K/AKT signaling in α′β′ MBn mediates HAB in sated flies, whereas Ras/ERK signaling in α′β′ MBn mediates HAB in hungry flies. Finally, we identified 2 α′β′ MBn downstream circuits, the MB output neuron (MBON)-α′3 and MBON-β′1, in which neuronal activity is required for HAB execution. Our results demonstrate that distinct Dilp signals mediate α′β′ MBn activity for proper HAB under different feeding states in *Drosophila*.

## Results

### MB activity regulates HAB

*D. melanogaster* is a small ectotherm whose body temperature is close to the temperature of its surroundings. *Drosophila* avoid extremely hot and cold environments and choose appropriate surrounding temperature (approximately 25˚C) for habitation [4–6]. We asked whether the internal feeding state affects the selection of appropriate surrounding temperature in flies. We used a thermoelectric device for our temperature preference behavioral analysis (Figs 1A and S1A–S1C, and S1 Video). To verify the accuracy and stability of the thermoelectric device, we measured the temperature in each quadrant of the plate using temperature sensors for 1 h at 25˚C and at various test temperatures (15˚C, 17˚C, 19˚C, 21˚C, 23˚C, 27˚C, 29˚C, 31˚C, 33˚C, and 35˚C). The margin of error on each aluminum plate was less than 1˚C (approximately 0.5˚C) at all test temperature settings (S1D Fig). Using this thermoelectric device to perform the two-choice assay [6,7], we found that hungry flies prefer to stay at a surrounding temperature of approximately 23˚C rather than 25˚C (Fig 1B), which is consistent with the results of a previous study showing that starvation reduces $T_p$ in *Drosophila* [18]. Interestingly, we observed an increased hot avoidance by hungry flies compared to food-sated flies; however, no differences were observed in the cold avoidance behavior (Fig 1B). It has been shown that mALT projection neurons convey hot stimuli from the antenna lobe to the calyx of the MB [7], and a recent study also suggests that hunger/satiety signals modulate the MB circuits [25], implying that MB is the integrative center for hot and hunger/satiety signals. Next, we investigated whether silencing or activating the MBn activity affects HAB. Constitutive silencing of the MBn activity by expressing the inward rectifier potassium channel Kir2.1 via *VT30559-GAL4* reduced HAB in both hungry and sated states (S2A and S3A Figs). To avoid the effects of constitutive Kir2.1 expression on neuronal development, we used optogenetic tools for temporal silencing or activation of MBn. Temporal silencing of the MBn activity by the blue light gated anion channel *GtACR2* [27] reduced HAB (Fig 1C), whereas temporal activation of MBn by the red light gated cation channel *CsChrimson* [28] increased HAB in both hungry and sated states (Fig 1D). Three min of blue or red light irradiation did not alter the setting temperatures on the aluminum plates of the thermoelectric device (S1E and S1F Fig). Resultantly, temporal blue or red light irradiation slightly shifted the HAB (S2B and S2C Fig) in comparison with the HAB without light irradiation (Fig 1B), indicating that exposure to light may result in a shift the set point and a switch in the valence of the hot stimulus. However, the differences of HAB between sated and hungry flies still exist under blue or red light treatment conditions (S2B and S2C Fig). These results confirm that MBn activity is positively correlated with *Drosophila* HAB in both hungry and sated states.

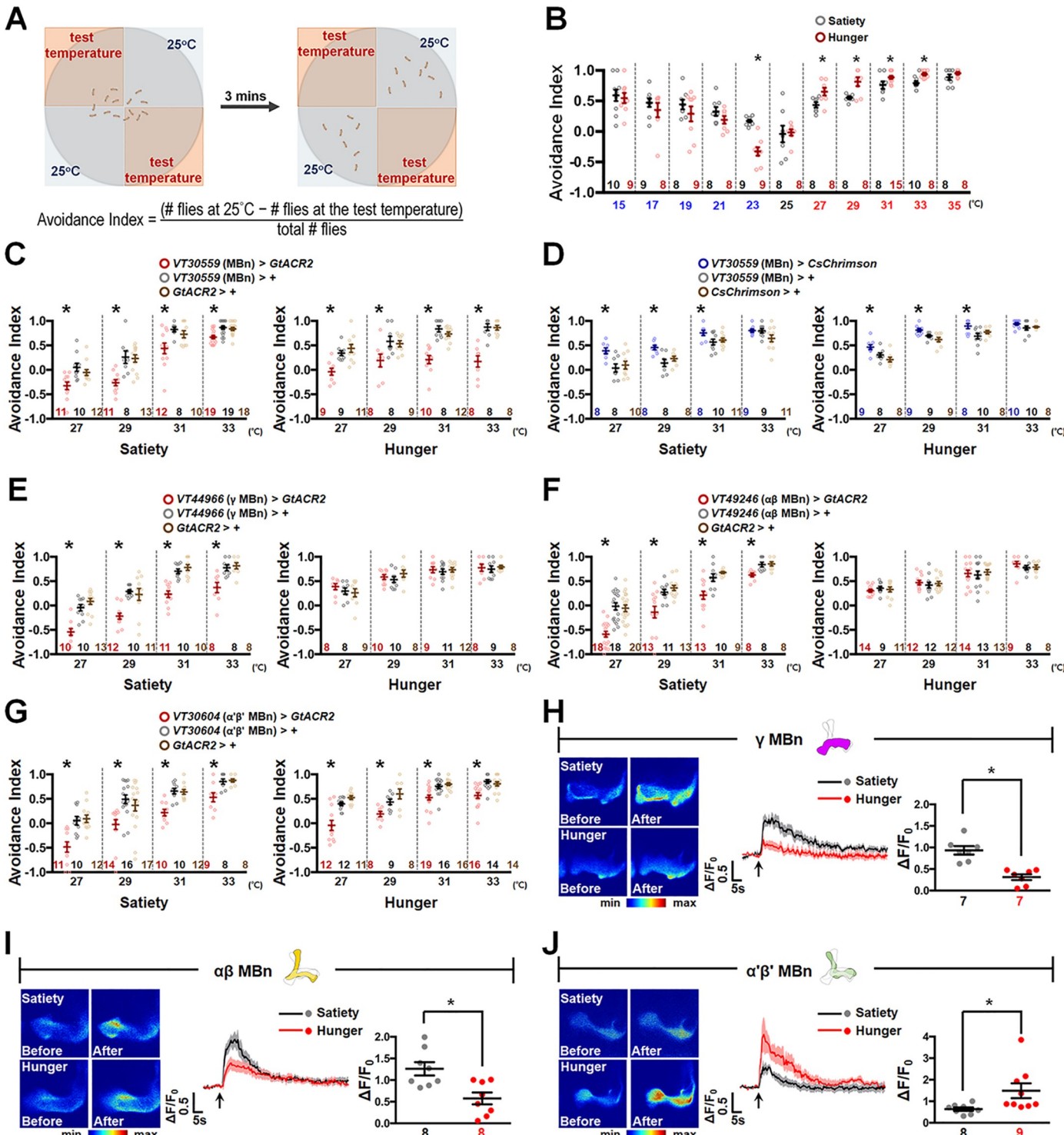

**Fig 1. α'β' MBn is required for HAB in both sated and hungry states.** (**A**) A diagram illustrating the two-choice assay experiment. (**B**) The aversion to temperature in the 15–35°C range during satiety (black) and hunger (red). Hungry flies showed significantly increased aversion to temperature in the 27–33°C range as compared to food-sated flies and preferred a temperature of 23°C (P-values: 0.7333, 0.3746, 0.3269, 0.1795, 0.0066, 0.8651, 0.0134, 0.0034, 0.0297, 0.0029, and 0.2309 from left to right). (**C**) Optogenetic silencing of MBn activity by *GtACR2* inhibits HAB in both feeding states (Satiety: P-values: 0.0032, 0.0002, 0.0119, and <0.0001 from left to right; Hunger: P-values: 0.0002, 0.0244, <0.0001, and <0.0001 from left to right). (**D**) Optogenetic activation of MBn activity by *CsChrimson* increases HAB in both sated and hungry states (Satiety: P-values: 0.0092, 0.0023, 0.049, and 0.0683 from left to right. Hunger: P-values: 0.0025, 0.0024, 0.0232, and 0.1536 from left to right). (**E**) Optogenetic silencing of γ MBn activity inhibited HAB in the sated state but not in the hungry state (Satiety: P-values:

<0.0001, 0.0002, <0.0001, and 0.001 from left to right; Hunger: *P*-values: 0.4572, 0.3521, 0.8352, and 0.855 from left to right). (**F**) Optogenetic silencing of αβ MBn activity inhibited HAB in the sated state but not in the hungry state (Satiety: *P*-values: < 0.0001, 0.0003, 0.0002, and 0.0019 from left to right; Hunger: *P*-values: 0.3912, 0.7685, 0.8361, and 0.4902 from left to right). (**G**) Optogenetic silencing of α′β′ MBn activity inhibits HAB in both states (Satiety: *P*-values: <0.0001, 0.0034, <0.0001, and 0.0019 from left to right; Hunger: *P*-values: <0.0001, 0.0027, <0.0001, and 0.0002 from left to right). (**H**) Hot stimuli induced the calcium response in γ MBn during satiety, while during hunger, the hot response was diminished ($P = 0.0002$). The GCaMP intensity changes ($\Delta F/F_0$) in MB γ lobe were recorded and analyzed. (**I**) Hot stimuli induced the calcium response in αβ MBn in the sated state, while in hungry state, this hot response was diminished ($P = 0.005$). The GCaMP intensity changes ($\Delta F/F_0$) in MB β lobe were recorded and analyzed. (**J**) Hot stimuli induced the calcium response in α′β′ MBn during satiety, whereas this hot response was enhanced during hunger ($P = 0.0373$). The GCaMP intensity changes ($\Delta F/F_0$) in MB β′ lobe were recorded and analyzed. The arrows under each calcium response curve indicate the time points at which the hot stimulus was applied. Each *N* represents either a group of 15 flies analyzed together in behavioral assays (**B–G**) or a single fly in live brain calcium imaging experiments (**H–J**). Data are represented as mean ± SEM with dots representing individual values. The data underlying this figure can be found in S1 Data. Data were analyzed by one-way ANOVA followed by Tukey's test (**C–G**) or the unpaired two-tailed *t* test (**B, H–J**). *$P < 0.05$. HAB, hot avoidance behavior; MBn, mushroom body neuron; SEM, standard error of mean.

## α′β′ MBn responds to hot stimuli and is required for HAB

Since MBn can be further classified into γ, αβ, and α′β′ neurons, we further investigated the role of each MBn subtype in regulating HAB. In food-sated flies, temporal silencing of each MBn subtype via *GtACR2* inhibited HAB (Figs 1E–1G, S2D–S2F and S3B–S3G). In the hungry state, *GtACR2*-mediated silencing of the α′β′ MBn activity inhibited HAB (Figs 1G and S2F); however, silencing of αβ or γ MBn activity had no effect (Figs 1E, 1F, S2D and S2E). Next, we investigated whether hot stimulus affects the activity of MBn subtypes. To visualize the neuronal activity, we genetically expressed *GCaMP7s* in each MBn and recorded calcium responses before and after the hot stimulus. Results showed significantly increased calcium responses in γ, αβ, and α′β′ MBn following hot stimuli in sated flies (Fig 1H–1J). Interestingly, in hungry flies, calcium responses in γ and αβ MBn were significantly lower, while the calcium response in α′β′ MBn was significantly higher compared to sated flies (Fig 1H–1J). Room temperature stimuli did not induce any significant calcium responses in γ, αβ, and α′β′ MBn, suggesting that these calcium responses were indeed induced by hot stimuli and were not movement artifacts of our live brain imaging operations (S2H Fig). Since temporal activation of MBn increased HAB in flies (Fig 1D), we asked whether activating α′β′ MBn specifically increases HAB in both hungry and sated states. Optogenetic activation of α′β′ MBn in flies expressing the *CsChrimson* transgene increased HAB in both feeding states (S2G Fig). These results are consistent with those of our behavioral study showing that only α′β′ MBn activity is required for HAB in sated and hungry flies (Fig 1G).

## InR mediates the hot response by inhibiting α′β′ MBn activity

Since Dilps play a role in satiety/hunger modulation [16], we asked whether Dilp signaling affects the hot response under different feeding states. Considering that *Drosophila* has 8 Dilps but only 1 InR [11], we first investigated the role of InR in MBn for HAB. Genetic expression of the dominant negative form of InR (*InR^DN*) in MBn increased HAB (Fig 2A), whereas expression of the constitutively active form of InR (*InR^CA*) in MBn decreased HAB (Fig 2B). Therefore, Dilp-induced InR signaling in MB may inhibit HAB in both feeding states. These behavioral changes were also observed when *InR^DN* and *InR^CA* were expressed in α′β′ MBn specifically (Figs 2C, 2D, S4A and S5A) and not in γ and αβ MBn (S4B–S4E Fig), indicating that insulin signaling specifically in α′β′ MBn is required for HAB. The gross morphology of α′β′ MBn was not affected in flies carrying *InR^DN* or *InR^CA* transgenes (S5D Fig), suggesting that the expression of *InR^DN* or *InR^CA* does not affect the development of α′β′ MBn. To completely exclude the developmental effect of *InR^DN* and *InR^CA* expression, we co-expressed the *tub-GAL80^ts* transgene for the acute control of *InR^DN* and *InR^CA* expression. Similar behavioral phenotypes were observed in flies with acute expression of *InR^DN* or *InR^CA* in α′β′ MBn (Figs 2E, 2F, S4F and S5B). Furthermore, acute expression of *InR^DN* or *InR^CA* in α′β′ MBn had

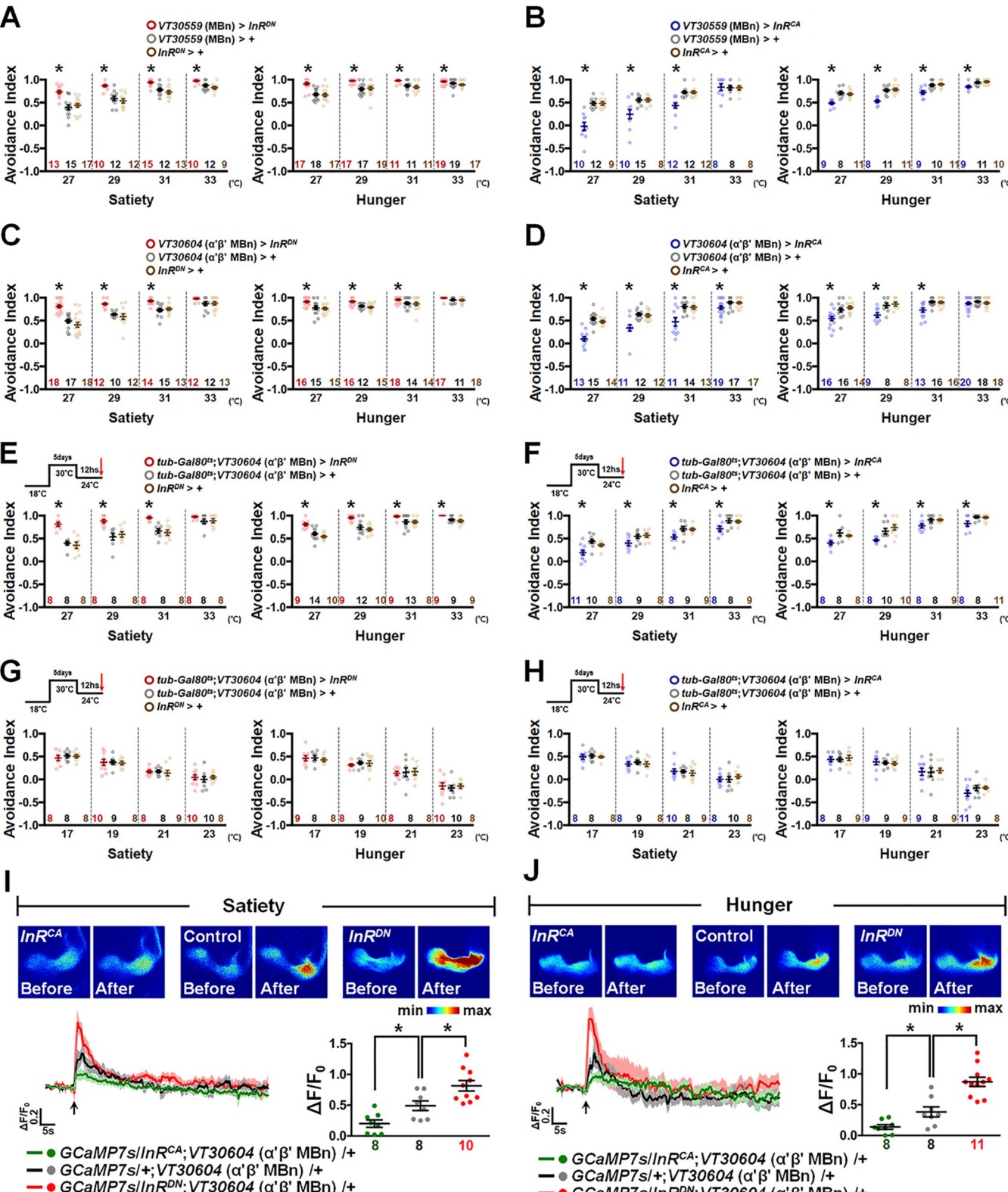

**Fig 2. Insulin signaling in α'β' MBn is critical for HAB.** (**A**) Genetic expression of the dominant negative form of InR (*InR^DN*) in MBn increased HAB in both sated and hungry states (Satiety: *P*-values: <0.0001, <0.0001, <0.0001, and 0.0055 from left to right; Hunger: *P*-values: <0.0001, <0.0001, 0.0026, and 0.0291 from left to right). (**B**) Genetic expression of the constitutive active form of InR (*InR^CA*) in MBn decreased HAB in both sated and hungry states (Satiety:

*P*-values: <0.0001, 0.0035, <0.0001, and 0.485 from left to right; Hunger: *P*-values: <0.0001, <0.0001, 0.0005, and 0.0022 from left to right). (**C**) Genetic expression of *InR^DN* in α′β′ MBn increased HAB in both sated and hungry states (Satiety: *P*-values: <0.0001, <0.0001, <0.0001, and 0.0506 from left to right; Hunger: *P*-values: 0.0016, <0.0001, 0.0309, and 0.0562 from left to right). (**D**) Genetic expression of *InR^CA* in α′β′ MBn decreased HAB in both sated and hungry states (Satiety: *P*-values: <0.0001, 0.0002, 0.0004, and 0.0138 from left to right; Hunger: *P*-values: 0.0003, 0.0043, 0.0003, and 0.4695 from left to right). (**E**) Adult-stage-specific expression of *InR^DN* in α′β′ MBn increased HAB in both sated and hungry states (Satiety: *P*-values: <0.0001, 0.0012, <0.0001, and 0.2074 from left to right; Hunger: *P*-values: <0.0001, 0.0002, 0.0418, and 0.0035 from left to right). (**F**) Adult-stage-specific expression of *InR^CA* in α′β′ MBn decreased HAB in both sated and hungry states (Satiety: *P*-values: 0.0013, 0.0379, 0.008, and 0.0198 from left to right; Hunger: *P*-values: 0.0094, 0.0051, 0.0246, and 0.0155 from left to right). (**G**) Adult-stage-specific expression of *InR^DN* in α′β′ MBn did not affect cold avoidance behavior in both sated and hungry states (Satiety: *P*-values: 0.7397, 0.9641, 0.8158, and 0.7703 from left to right; Hunger: *P*-values: 0.803, 0.7378, 0.9292, and 0.8924 from left to right). (**H**) Adult-stage-specific expression of *InR^CA* in α′β′ MBn did not affect cold avoidance behavior in both sated and hungry states (Satiety: *P*-values: 0.908, 0.7472, 0.8053, and 0.6225 from left to right; Hunger: *P*-values: 0.8746, 0.8515, 0.9402, and 0.2481 from left to right). (**I, J**) Genetic expression of *InR^CA* decreased hot-induced calcium response, whereas expression of *InR^DN* increased hot-induced calcium response in α′β′ MBn compared to control groups in sated (**I**) and hungry (**J**) flies (*P* = 0.0145 and 0.0102 for satiety; *P* = 0.0004 and 0.0168 for hunger). The arrows under each calcium response curve indicate the time points at which the hot stimulus was applied. The GCaMP intensity changes ($\Delta F/F_0$) in MB β′ lobe were recorded and analyzed in each calcium imaging data. Each *N* represents either a group of 15 flies analyzed together in the behavioral assay (**A**–**H**) or a single fly in calcium imaging experiments (**I, J**). Data are represented as mean ± SEM with dots representing individual values. The data underlying this figure can be found in S1 Data. Data were analyzed by one-way ANOVA followed by Tukey's test (**A**–**H**) or the unpaired two-tailed *t* test (**I, J**). *\*P* < 0.05. HAB, hot avoidance behavior; MBn, mushroom body neuron; SEM, standard error of mean.

no effect on the cold avoidance behavior, suggesting that Dilp signaling in MB mediates hot but not cold responses (Figs 2G, 2H, S4G, S4H and S5C). Live brain imaging data showed that *InR^DN* transgene expression enhanced the hot stimuli-induced calcium response, whereas *InR^CA* expression reduced the calcium response in α′β′ MBn in both sated (Fig 2I) and hungry (Fig 2J) flies. Together, our data suggest that InR negatively regulates α′β′ MBn activity, which contributes to HAB in both sated and hungry states.

## Dilp2 mediates α′β′ MBn activity required for HAB during satiety via PI3K/AKT signaling

In the fly brain, Dilp2, Dilp3, and Dilp5 are released from IPCs during satiety; therefore, we investigated HAB in *dilp* mutant flies. Behavioral screening showed significantly increased HAB in sated *dilp2* but not *dilp3* and *dilp5* mutant flies (Figs 3A and S6A). Since α′β′ MBn is critical for the hot response, we investigated whether loss of Dilp2 affects α′β′ MBn response to the hot stimulus. Live brain imaging data showed an increased calcium response to the hot stimulus in α′β′ MBn of *dilp2* mutant flies in the sated state (Fig 3B, left panel). However, the increased calcium response in α′β′ MBn was not observed in hungry *dilp2* mutant flies, further suggesting that Dilp2 mediates hot responses only during satiety (Fig 3B, right panel). Since Dilp2 is secreted by IPCs in the brain, we investigated HAB in IPCs-specific *dilp2* knockdown flies (S6B Fig). To exclude the developmental effect of decrease IPCs-specific *dilp2* expression, we co-expressed the *tub-GAL80^ts* transgene for adult-stage-specific silencing of *dilp2* in IPCs increased HAB in the sated but not in the hungry state (Figs 3C and S6C), suggesting that Dilp2 secreted by IPCs inhibits HAB only during satiety. Our immunohistochemistry data also showed abundant accumulation of Dilp2 in IPCs during the hungry state (S6D Fig), which is consistent with the results of previous studies in *Drosophila* larvae [16] and adult flies [17]. We performed live brain imaging to determine whether the Dilp2 secretion by IPCs represent satiety or a hot signal. Live brain imaging data showed that IPCs are not responsive to hot stimuli, which indicates that Dilp2 secretion represents the sated state rather than the hot signal (S6E Fig).

To assess whether Dilp2 produced by IPCs inhibits the hot response of α′β′ MBn, we genetically silenced *dilp2* in IPCs and recorded the calcium response in α′β′ MBn before and after the hot stimulus. Results showed that silencing *dilp2* in IPCs significantly increased the calcium response to hot stimulus in α′β′ MBn in sated but not in hungry flies (Fig 3D). Furthermore, *dilp2* overexpression in IPCs inhibited HAB (Figs 3E and S6F), and live brain imaging

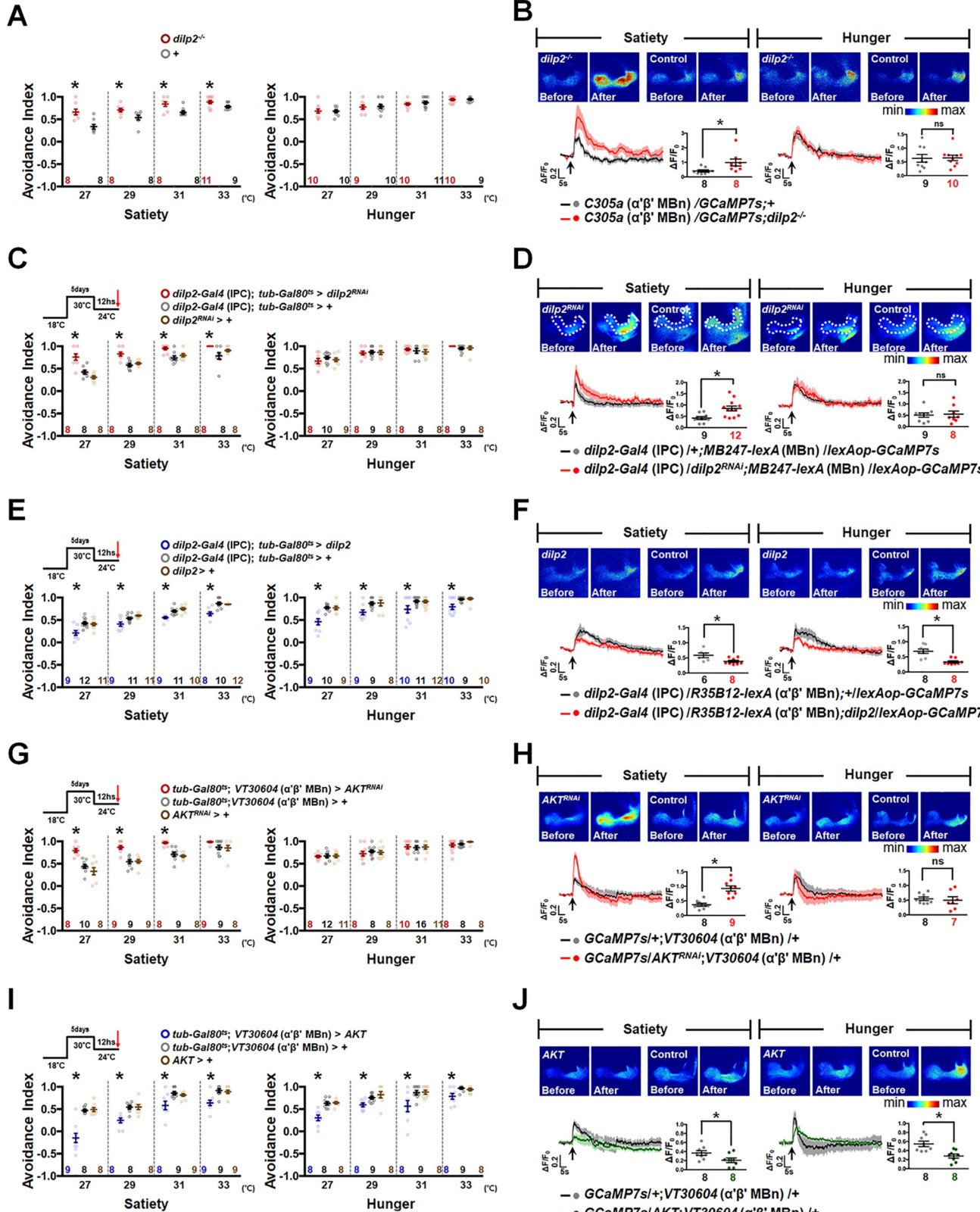

**Fig 3. Dilp2 secreted by IPCs regulates AKT signaling for HAB during satiety.** (A) *dilp2* mutant flies showed increased HAB during satiety but not during hunger (Satiety: *P*-values: 0.0013, 0.0187, 0.01, and 0.0412 from left to right; Hunger: *P*-values: 0.9355, 0.8074, 0.5071, and 0.8233 from left to right).

(**B**) α′β′ MBn showed increased hot response in *dilp2* mutant background specifically in sated but not in hungry state (*P* = 0.0394 for satiety; *P* = 0.9146 for hunger). (**C**) Adult-stage-specific knockdown of *dilp2* in IPCs increased HAB during satiety but not during hunger (Satiety: *P*-values: <0.0001, 0.0001, 0.0026, and 0.0115 from left to right; Hunger: *P*-values: 0.4499, 0.9351, 0.695, and 0.4026 from left to right). (**D**) Genetic knockdown of *dilp2* in IPCs enhanced hot response in α′β′ MBn specifically during satiety (*P* = 0.0057 for satiety; *P* = 0.8046 for hunger). (**E**) Adult-stage-specific expression of *dilp2* in IPCs decreases HAB during both satiety and hunger (Satiety: *P*-values: 0.0006, 0.0005, <0.0001, and <0.0001 from left to right; Hunger: *P*-values: <0.0001, 0.0134, 0.0286, and 0.0039 from left to right). (**F**) Genetic expression of *dilp2* in IPCs decreased hot response of α′β′ MBn during both satiety and hunger (*P* = 0.0238 for satiety; *P* = 0.0005 for hunger). (**G**) Adult-stage-specific knockdown of *AKT* in α′β′ MBn increased HAB during satiety (Satiety: *P*-values: <0.0001, <0.0001, <0.0001, and 0.0766 from left to right; Hunger: *P*-values: 0.5511, 0.8163, 0.5558, and 0.3186 from left to right). (**H**) Genetic knockdown of *AKT* in α′β′ MBn increased hot responses of α′β′ MBn during satiety but not during hunger (*P* = 0.0001 for satiety; *P* = 0.7126 for hunger). (**I**) AKT overexpression in α′β′ MBn decreased HAB in both sated and hungry states (Satiety: *P*-values: <0.0001, 0.0004, 0.0057, and 0.0004 from left to right; Hunger: *P*-value: <0.0001, 0.0164, 0.0161, and 0.0117 from left to right). (**J**) AKT overexpression in α′β′ MBn decreased hot response of α′β′ MBn in both sated and hungry states (*P* = 0.0453 for satiety; *P* = 0.0035 for hunger). The arrows under each calcium response curve indicate the time points at which the hot stimulus was applied. The GCaMP intensity changes (ΔF/F_0) in MB β′ lobe were recorded and analyzed in each calcium imaging data. Each *N* represents either a group of 15 flies analyzed together in the behavioral assay (**A, C, E, G, I**) or a single fly in calcium imaging experiments (**B, D, F, H, J**). Data are represented as mean ± SEM with dots representing individual values. The data underlying this figure can be found in S1 Data. Data were analyzed by one-way ANOVA followed by Tukey's test (**C, E, G, I**) or the unpaired two-tailed *t* test (**A, B, D, F, H, J**). \**P* < 0.05; ns, not significant. HAB, hot avoidance behavior; IPC, insulin-producing cell; MBn, mushroom body neuron; SEM, standard error of mean.

data also showed decreased calcium responses to hot stimuli in α′β′ MBn (Fig 3F). These results support the notion that Dilp2 secreted by IPCs suppresses the hot response and HAB by inhibiting α′β′ MBn activity. Dilps-InR interaction induces multiple signaling pathways, including PI3K/AKT and Ras/ERK pathways [29–31]. We next examined the potential downstream components of InR in α′β′ MBn that are required for HAB. In the sated state, we found that constitutive expression of the dominant negative PI3K (*PI3K^DN*) (S7A and S7B Fig) or RNAi-mediated knockdown of *AKT* (S7C and S7D Fig) in α′β′ MBn increased HAB. To exclude the potential developmental effects of these manipulations, we introduced the *tub-Gal80^ts* transgene and showed that adult-stage-specific expression of *PI3K^DN* (S7E and S7F Fig) or knockdown of *AKT* (Figs 3G and S7G) in α′β′ MBn increased HAB in the sated but not in the hungry state. Furthermore, *AKT* knockdown in α′β′ MBn increased the calcium response to hot stimuli in the sated state (Fig 3H). These results suggest that Dilp2 secreted by IPCs induces PI3K/AKT signaling in α′β′ MBn, which contributes to HAB and hot responses during satiety. Our results showed that constitutive expression of AKT (S7H Fig) or adult-stage-specific *AKT* expression (Figs 3I and S7I) in α′β′ MBn reduced HAB. Live brain imaging data showed that AKT overexpression in α′β′ MBn decreased the hot response (Fig 3J). Taken together, our data imply that PI3K/AKT signaling inhibits HAB and hot responses by suppressing hot stimuli-induced α′β′ MBn activity.

## Dilp6 modulates α′β′ MBn activity for HAB in hungry flies via Ras/ERK signaling

The genetic silencing of *dilp2* in IPCs or the inhibition of PI3K/AKT signaling in α′β′ MBn increased HAB in food-sated but not in hungry flies (Figs 3C, 3G, S6B, S6C and S7A–S7G). However, manipulating InR expression in the MB affects HAB in both sated and hungry states (Fig 2), suggesting the existence of signals other than Dilp2 for HAB modulation during the hungry state. It has been shown that Dilp6 is secreted by the fat body during starvation [19], which prompted us to examine whether manipulating Dilp6 expression affects HAB. We observed that *dilp6* loss-of-function mutant flies (*Dilp6^LOF*) exhibited increased HAB when hungry but not when sated (Fig 4A). In addition, the calcium response to the hot stimulus in α′β′ MBn was significantly increased only in hungry *Dilp6^LOF* flies (Fig 4B). Constitutive knockdown of *dilp6* in the fat body via *cg-GAL4* increased HAB in hungry but not in the sated state (S8A Fig). Moreover, adult-stage-specific knockdown of *dilp6* in the fat body increased HAB (Figs 4C and S8B) and enhanced the hot response (Fig 4D) in α′β′ MBn, specifically in hungry flies. Conversely, adult-stage-specific overexpression of *dilp6* in the fat body reduced

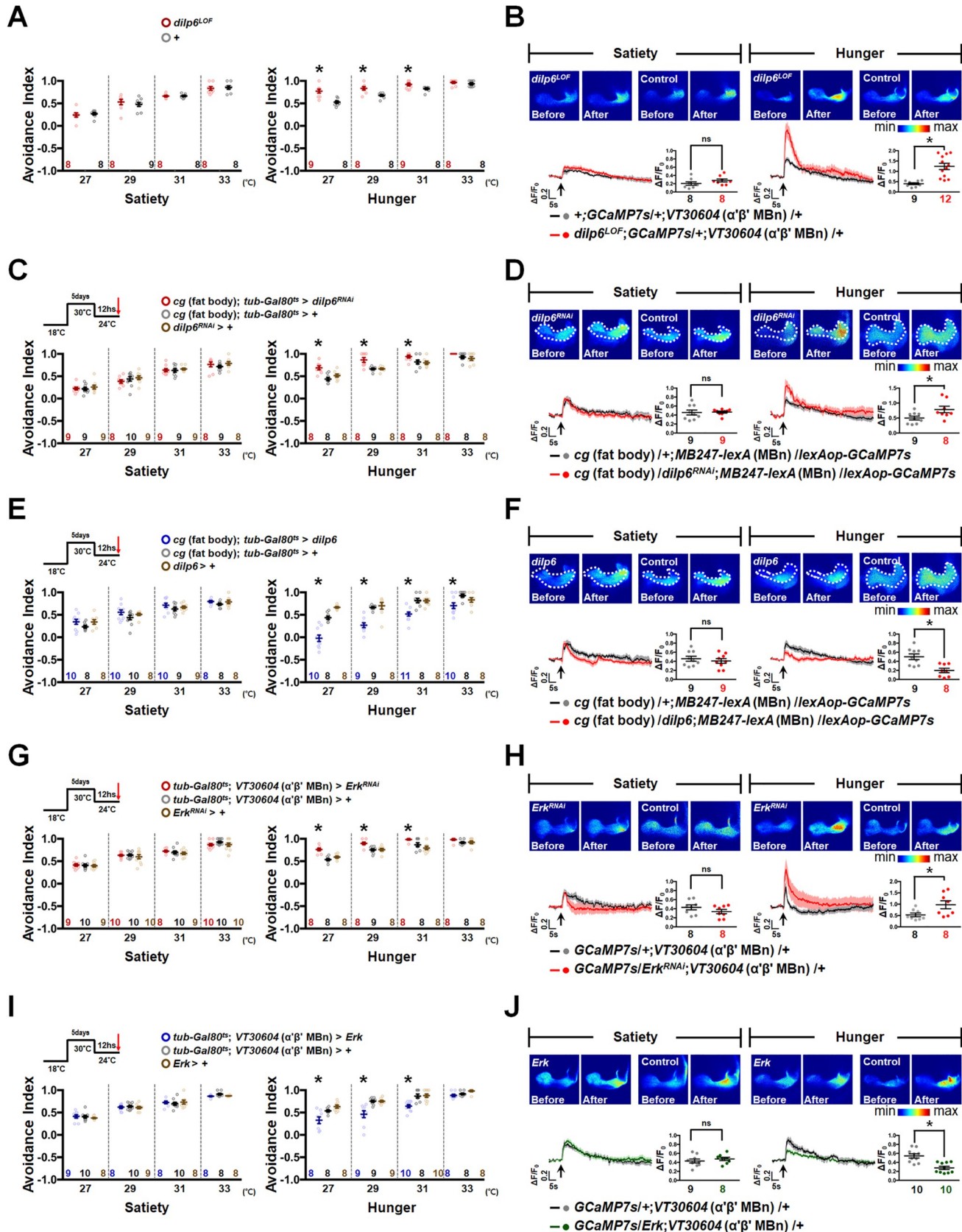

**Fig 4. Dilp6 produced by the fat body regulates ERK for HAB during the hungry state.** (**A**) *dilp6* loss-of-function mutant flies (*dilp6*<sup>LOF</sup>) showed increased HAB when hungry (Satiety: *P*-values: 0.6176, 0.4785, 0.8634, and 0.7292 from left to right; Hunger: *P*-values: 0.0003, 0.0029, 0.0222, and 0.4018 from left to right). (**B**) α′β′ MBn showed increased hot response in *dilp6*<sup>LOF</sup> mutant background specifically when hungry but not when sated (*P* = 0.2043 for satiety; *P* = 0.0002 for hunger). (**C**) Adult-stage-specific knockdown of *dilp6* in the fat body increased HAB in hungry but not in sated flies (Satiety: *P*-values: 0.7112, 0.4332, 0.748, and 0.4753 from left to right; Hunger: *P*-values: 0.0006, 0.0002, 0.0332, and 0.0732 from left to right). (**D**) Genetic knockdown of *dilp6* in the fat body increased hot response of α′β′ MBn specifically in the hungry state (*P* = 0.8682 for satiety; *P* = 0.0328 for hunger). (**E**) Adult-stage-specific expression of *dilp6* in the fat body decreased HAB in the hungry state (Satiety: *P*-values: 0.2437, 0.2066, 0.3148, and 0.2978 from left to right; Hunger: *P*-values: <0.0001, <0.0001, <0.0001, and 0.0194 from left to right). (**F**) Genetic expression of *dilp6* in the fat body decreased hot response of α′β′ MBn in the hungry state (*P* = 0.5214 for satiety; *P* = 0.002 for hunger). (**G**) Adult-stage-specific knockdown of *Erk* in α′β′ MBn increased HAB during hunger (Satiety: *P*-values: 0.9053, 0.6805, 0.4785, and 0.2593 from left to right; Hunger: *P*-values: <0.0001, 0.0011, 0.0026, and 0.108 from left to right). (**H**) Genetic knockdown of *Erk* in α′β′ MBn increased hot responses of α′β′ MBn during hunger but not during satiety (*P* = 0.2155 for satiety; *P* = 0.0361 for hunger). (**I**) Adult-stage-specific expression of *Erk* in α′β′ MBn decreased HAB during the hungry state (Satiety: *P*-values: 0.6868, 0.7727, 0.7495, and 0.0552 from left to right; Hunger: *P*-values: 0.001, 0.0003, 0.0002, and 0.0121 from left to right). (**J**) Genetic expression of *Erk* in α′β′ MBn decreased the hot response of α′β′ MBn in the hungry state (*P* = 0.4394 for satiety; *P* = 0.0003 for hunger). The arrows under each calcium response curve indicate the time points at which the hot stimulus was applied. Each *N* represents either a group of 15 flies analyzed together in the behavioral assay (A, C, E, G, I) or a single fly in calcium imaging experiments (B, D, F, H, J). Data are represented as mean ± SEM with dots representing individual values. The data underlying this figure can be found in S1 Data. Data were analyzed by one-way ANOVA followed by Tukey's test (C, E, G, I) or the unpaired two-tailed *t* test (A, B, D, F, H, J). *P* < 0.05; ns, not significant. HAB, hot avoidance behavior; MBn, mushroom body neuron; SEM, standard error of mean.

HAB in hungry flies (Figs 4E and S8C), and live calcium imaging data also support the notion that *dilp6* reduces the hot response in α′β′ MBn during the hungry state and not during the sated state (Fig 4F). It might be possible that Dilp2 strongly inhibits α′β′ MBn activity in the sated state; therefore, Dilp6 overexpression failed to further reduce the neuronal activity in this state (Figs 1J, 3F and 4F).

Blocking the PI3K/AKT signaling in α′β′ MBn increased HAB in sated flies; however, it had no effect on HAB in hungry flies (Figs 3G and S7A–S7G). There are 2 major cellular signaling pathways, PI3K/AKT and Ras/ERK pathways, which can be activated by InR [29–31]. We asked whether other Dilp-InR signals regulate HAB and α′β′ MBn activity during the hungry state. Intriguingly, RNAi-mediated knockdown of *Ras*, *Raf*, or *Erk* in α′β′ MBn induced a strong HAB only in the hungry state but not in the sated state (S9A–S9F Fig). Similarly, adult-stage-specific knockdown of *Ras*, *Raf*, or *Erk* in α′β′ MBn induced a strong HAB, specifically in the hungry state (Figs 4G and S9G–S9K). This phenotype is similar to that observed in *dilp6* loss-of-function mutants (Fig 4A) and flies with *dilp6* knockdown in the fat body (Fig 4C). Live calcium imaging data also support the notion that RNAi-mediated *Erk* knockdown in α′β′ MBn increased hot response only in hungry but not in sated flies (Fig 4H). Constitutive or adult-stage-specific expression of *Erk* in α′β′ MBn reduced HAB (Figs 4I, S9L and S9M) and the hot response of α′β′ MBn (Fig 4J) in hungry flies, which supports the notion that Dilp6/ERK signaling inhibits the hot stimulus-induced α′β′ MBn activity specifically in the hungry state.

Considering that the phosphorylation of AKT and ERK represents the activity of PI3K/AKT and Ras/ERK signaling pathways [30,32], we asked whether the expression of phospho-AKT (pAKT) and phospho-ERK (pERK) in α′β′ MBn in sated flies differs from that in hungry flies. Immunohistochemical analysis revealed higher expression levels of pAKT in the sated state (Fig 5A), whereas higher expression levels of pERK in hungry flies were observed (Fig 5B). However, the genetic expression of *InR*<sup>DN</sup> in α′β′ MBn ameliorated the increased levels of pAKT and pERK in sated and hungry flies, respectively (Fig 5C and 5D). The genetic silencing of *dilp2* in IPCs also suppressed the levels of pAKT in sated flies (Fig 5E), whereas silencing of *dilp6* in the fat body suppressed the levels of pERK in hungry flies (Fig 5F). Overall, our results suggest that satiety-induced Dilp2 secretion by IPCs triggers an intracellular signaling in α′β′ MBn, which is distinct from hunger-induced Dilp6 secretion from the fat body. These 2 distinct InR-dependent intracellular signals contribute to proper HAB under both feeding states in *Drosophila*.

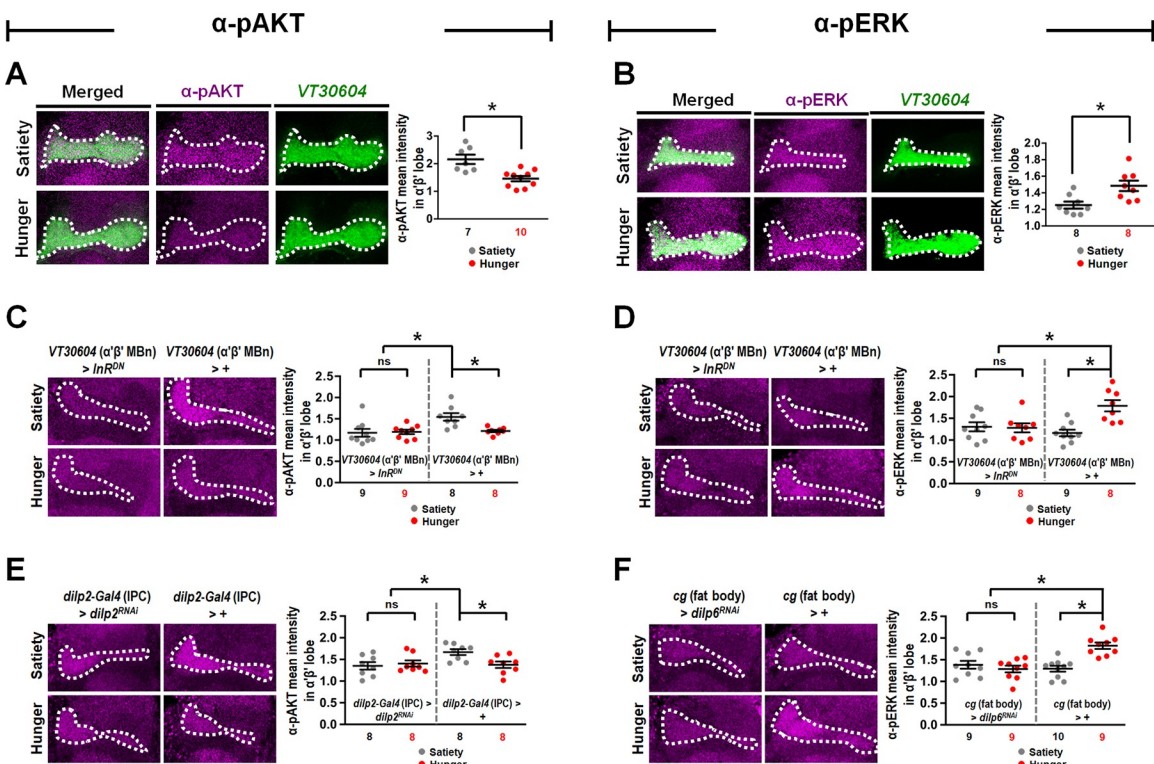

**Fig 5. Dilp2 increases pAKT while Dilp6 increases pERK levels in α′β′ MBn during satiety and hunger.** (**A**) Immunostaining with pAKT antibody in *VT30604-GAL4 > UAS-mCD8::GFP* flies. Quantification of anti-pAKT immunopositive signals in α′β′ MBn in sated and hungry states (right panel) ($P = 0.0014$). (**B**) Immunostaining with pERK antibody in *VT30604-GAL4 > UAS-mCD8::GFP* flies. Quantification of anti-pERK immunopositive signals in α′β′ MBn in sated and hungry states (right panel) ($P = 0.0083$). (**C**) Immunostaining with pAKT antibody in *VT30604-GAL4 > UAS-InR$^{DN}$* and *VT30604-GAL4 > +* flies. Quantification of anti-pAKT immunopositive signals in α′β′ MBn in sated and hungry states (right panel) ($P = 0.8402, 0.0047$, and $0.0031$ from left to right). (**D**) Immunostaining with pERK antibody in *VT30604-GAL4 > UAS-InR$^{DN}$* flies and *VT30604-GAL4 > +* flies. Quantification of anti-pERK immunopositive signals in α′β′ MBn in sated and hungry states (right panel) ($P = 0.8662, 0.0066$, and $0.0006$ from left to right). (**E**) Immunostaining with pAKT antibody in *dilp2-GAL4 > UAS-dilp2$^{RNAi}$* and *dilp2-GAL4 > +* flies. Quantification of anti-pAKT immunopositive signals in α′β′ MBn in sated and hungry states (right panel) ($P = 0.6434, 0.0158$, and $0.0109$ from left to right). (**F**) Immunostaining with pERK antibody in *cg-GAL4 > UAS-dilp6$^{RNAi}$* and *cg-GAL4 > +* flies. Quantification of anti-pERK immunopositive signals in α′β′ MBn in sated and hungry states (right panel) ($P = 0.4332, 0.0002$, and $<0.0001$ from left to right). Each *N* represents a single fly in anti-pAKT or anti-pERK immunostaining experiments. All the anti-pAKT and anti-pERK immunostaining signals in α′β′ MBn were normalized to the signals in subesophageal ganglion. Data are represented as mean ± SEM with dots representing individual values. The data underlying this figure can be found in S1 Data. Data were analyzed by the unpaired two-tailed *t* test (**A–F**) or one-way ANOVA followed by Tukey's test (**C–F**). \**P* < 0.05; ns, not significant. MBn, mushroom body neuron; SEM, standard error of mean.

## mALT conveys the hot signal to α′β′ MBn via the cholinergic transmission

The mALT projection neurons convey thermal stimuli from the antenna lobe to the MB, LH, and PLP regions in the fly brain [7]. We investigated whether α′β′ MBn responses to hot stimuli are delivered via mALT and the neurotransmitter involved in this process. To address this, we labeled the mALT neurons using *VT40053-GAL4* (Fig 6A) [7]. The axons of these neurons innervate the dendritic region of the MB, also known as calyx (Fig 6B). Immunostaining using the anti-choline acetyltransferase (anti-ChAT) antibody showed positive signals in mALT axons, suggesting that these neurons use acetylcholine for neurotransmission (Figs 6C and S10A) [33]. Furthermore, *kir2.1* transgene expression in mALT neurons reduced HAB in both sated and hungry states, suggesting that mALT neuronal activity is required for normal HAB (S10B Fig). To avoid the developmental effects of *kir2.1* transgene expression in neurons,

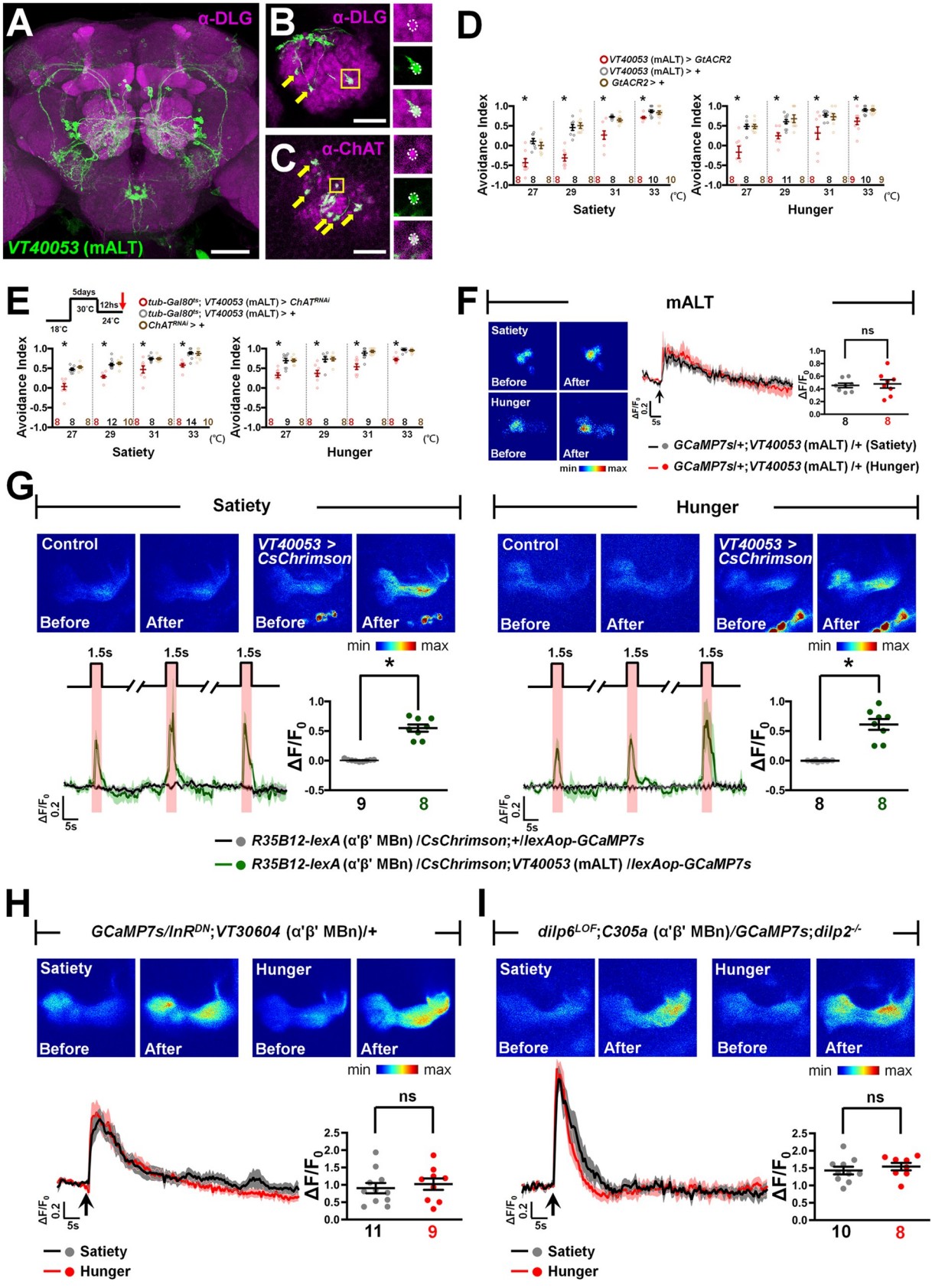

**Fig 6. Cholinergic mALT neurons convey hot information to α′β′ MBn.** (**A**) The morphology of mALT neurons in the fly brain. Brain is counterstained with anti-DLG antibody. (**B**) The synaptic bouton-like structure (arrows) of mALT in the MB calyx region. Brain is counterstained with anti-DLG antibody. (**C**) The co-localization (arrows) of anti-ChAT immunostaining signals (magenta) and mALT axons (green). (**D**) Optogenetic silencing of mALT activity via *GtACR2* inhibited HAB in both hungry and sated states (Satiety: *P*-values: 0.0003, <0.0001, 0.0001, and 0.0271 from left to right; Hunger: *P*-values: 0.0001, 0.001, 0.0103, and 0.001 from left to right). (**E**) Adult-stage-specific knockdown of ChAT in mALT inhibited HAB during both feeding states (Satiety: *P*-values: <0.0001, <0.0001, 0.0029, and <0.0001 from left to right; Hunger: *P*-values: <0.0001, 0.0005, <0.001, and <0.0001 from left to right). (**F**) Hot stimulus evoked the calcium response in hot glomerulus of mALT and this calcium response was not significantly different between sated (black) and hungry (red) flies (*P* = 0.7554). The GCaMP intensity changes ($\Delta F/F_0$) in mALT dendrites were recorded and analyzed. (**G**) Optogenetic activation of mALT via CsChrimson evoked calcium responses in α′β′ MBn in flies carry *R35B12-lexA/CsChrimson*; *VT40053/lexAop-GCaMP7* (green) but not in flies carrying *R35B12-lexA/CsChrimson*; *+/lexAop-GCaMP7* flies (black) (*P* < 0.0001 for satiety; *P* < 0.0001 for hunger). (**H**) The calcium responses to hot stimuli in α′β′ MBn do not differ significantly between sated (black) and hungry (red) flies when *InR^{DN}* transgene is expressed in α′β′ MBn (*P* = 0.6093). (**I**) The calcium responses to hot stimuli in α′β′ MBn do not differ significantly between sated (black) and hungry (red) *dilp2^{-/-}* and *dilp6^{LOF}* double mutant flies (*P* = 0.4850). The GCaMP intensity changes ($\Delta F/F_0$) in MB β′ lobe were recorded and analyzed. The arrows under each calcium response curve indicate the time points at which the hot stimulus was applied. Each *N* represents either a group of 15 flies analyzed together in the behavioral assay (**D**, **E**) or a single fly in calcium imaging experiments (**F–I**). Data are represented as mean ± SEM with dots representing individual values. The data underlying this figure can be found in S1 Data. Data were analyzed by one-way ANOVA followed by Tukey's test (**D**, **E**) or the unpaired two-tailed *t* test (**F–I**). **P* < 0.05; ns, not significant. HAB, hot avoidance behavior; mALT, medial antennal lobe tract; MBn, mushroom body neuron; SEM, standard error of mean.

we expressed the *GtACR2* transgene in mALT neurons for temporal silencing of neuronal activity through blue light irradiation. Again, our results showed that temporal silencing of mALT neuronal activity reduced HAB (Fig 6D). RNAi-mediated knockdown of *ChAT* in mALT neurons also reduced HAB (S10C Fig), and adult-stage-specific knockdown of *ChAT* showed similar effects (Figs 6E and S10D) suggesting that cholinergic transmission in mALT neurons is required for hot signal input in both sated and hungry states. Live calcium imaging revealed the same level of hot response in mALT neurons under sated and hungry states (Fig 6F), implying that the hot input from mALT neurons is not affected by different feeding states. Next, we investigated whether α′β′ MBn indeed receives inputs from the mALT neurons. We genetically expressed *CsChrimson* in mALT neurons via *VT40053-GAL4 > UAS-CsChrimson*, combined with *GCaMP7s* expression in α′β′ MBn via *R35B12-lexA > lexAop-GCaMP7s* in a single fly. Live brain imaging showed significantly increased calcium responses during red light irradiation in transgenic flies compared to the control flies (Fig 6G). Furthermore, hot stimuli failed to induce the calcium response in α′β′ MBn when mALT neuronal activity was suppressed by ectopic expression of the *kir2.1* transgene (S10E Fig).

Next, we asked whether α′β′ MBn shows similar hot responses in hunger and sated states when the InR signal is suppressed. We genetically expressed *InR^{DN}* in α′β′ MBn via *VT30604-GAL4* and performed live calcium brain imaging. Results showed that calcium responses to hot stimuli did not significantly differ between both feeding states (Fig 6H). Moreover, we also investigated the α′β′ MBn response to hot stimuli in *dilp2^{-/-}* and *dilp6^{LOF}* double mutant flies. We found that the calcium response to hot stimuli did not significantly differ during both feeding states (Fig 6I). Overall, these results indicate that cholinergic mALT neurons are the major input for hot stimuli in α′β′ MBn in both sated as well as hungry flies and that Dilp2 and Dilp6 are only involved in the suppression of α′β′ MBn activity under sated and hungry states, respectively (Figs 2–6).

## MBON-α′3 and MBON-β′1 are essential for HAB execution

Since the hot input is conveyed by cholinergic mALT to α′β′ MBn (Fig 6), whereas Dilp2 and Dilp6 regulate α′β′ MBn activity during sated and hungry states, respectively (Figs 3–5), we investigated the output neurons that are critical for HAB. The information present in the MB can be read out by specific MBONs, the dendrites of which are restricted to different domains

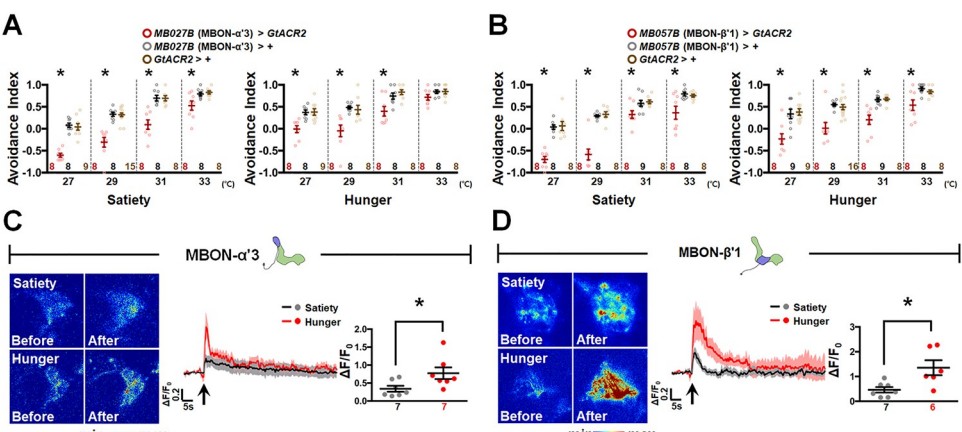

**Fig 7. MBON-α′3 and -β′1 output the converged information from α′β′ MBn.** (**A**) Optogenetic silencing of MBON-α′3 (labeled by *MB027B-GAL4*) activity inhibited HAB in both feeding states (Satiety: *P*-values: <0.0001, <0.0001, <0.0001, and 0.0109 from left to right; Hunger: *P*-values: 0.0008, 0.0016, 0.0026, and 0.1663 from left to right). (**B**) Optogenetic silencing of MBON-β′1 (labeled by *MB057B-GAL4*) activity inhibited HAB in both feeding states (Satiety: *P*-values: <0.0001, <0.0001, 0.0136, and 0.0053 from left to right; Hunger: *P*-values: 0.0004, 0.0001, <0.0001, and 0.0056 from left to right). (**C**) Hot stimulus induced the calcium response in MBON-α′3 in sated flies (black), whereas an increased hot response was observed in hungry flies (red) (*P* = 0.0353). The GCaMP intensity changes (ΔF/F$_0$) in MBON-α′3 dendrites were recorded and analyzed. (**D**) Hot stimulus induced the calcium response in MBON-β′1 in sated flies (black), whereas an increased hot response was observed in hungry flies (red) (*P* = 0.0134). The GCaMP intensity changes (ΔF/F$_0$) in MBON-β′1 dendrites were recorded and analyzed. The arrows under each calcium response curve indicate the time points at which the hot stimulus was applied. Each *N* represents either a group of 15 flies analyzed together in behavioral assays (**A**, **B**) or the response of a single fly in calcium imaging experiments (**C**, **D**). Data are represented as mean ± SEM with dots representing individual values. The data underlying this figure can be found in S1 Data. Data were analyzed by one-way ANOVA followed by Tukey's test (**A**, **B**) or the unpaired two-tailed *t* test (**C**, **D**). *P < 0.05. HAB, hot avoidance behavior; MBn, mushroom body neuron; MBON, mushroom body output neuron; SEM, standard error of mean.

of MB lobes [34]. It has been reported that the postsynaptic region of the MBON has high plasticity, and different MBONs encode distinct intrinsic valences, which can drive the approach or avoidance behavior in *Drosophila*. Therefore, summarizing valences in MBONs might be important for achieving a suitable behavioral output [34,35]. There are at least 9 different types of MBONs whose dendrites are restricted to distinct subdomains of α′β′ lobes [34,36] (S3H–S3P Fig). We individually tested the involvement of these α′β′-related MBONs in regulating HAB by *GtACR*-mediated silencing of neuronal activity (Figs 7A, 7B and S11). Our results showed that transient inhibition of MBON-α′3 (Fig 7A) or MBON-β′1 activity (Fig 7B) during behavioral assays disrupted HAB in both sated and hungry states. In addition, live brain imaging also indicated that MBON-α′3 and MBON-β′1 respond to hot stimuli in both feeding states (Fig 7C and 7D). These results suggest that the integrated information from α′β′ MBn is transmitted through MBON-α′3 and MBON-β′1 for proper HAB execution by sated and hungry flies.

## Discussion

In mice, it has been reported that insulin directly inhibits the activity of warm-sensitive neurons in the hypothalamus suggesting that insulin suppresses the sensation of hot stimuli [37]. The effects of insulin signaling on the physiological response to heat stress are now being explored. In this study, we revealed the cellular mechanisms underlying the regulation of insulin signaling for sensing hot temperatures under different feeding conditions in *Drosophila*. Flies prefer to stay at relatively higher temperatures in the food-sated state (Fig 1B). This differential response to the hot stimulus is tightly controlled by distinct insulin signals that produce

different levels of inhibition in α′β′ MBn (Fig 2). We demonstrated that flies exhibit a stronger HAB in the hungry state than in the sated state. Our data suggest that in the sated state, Dilp2 secreted by IPCs is responsible for conveying the satiety information to α′β′ MBn, which inhibits α′β′ MBn activity by inducing the PI3K/AKT signaling (Fig 3). In contrast, under hungry conditions, Dilp6 is released from the fat body to convey the hunger signal to α′β′ MBn, which inhibits α′β′ MBn activity by inducing the Ras/ERK signaling (Fig 4). Our immunohistochemistry data also support the notion that increased levels of pAKT and pERK in α′β′ MBn under sated and hungry states, respectively (Fig 5). Our results further suggest that Dilp2 increases pAKT levels while Dilp6 increases pERK levels in α′β′ MBn during sated and hungry states (Fig 5E and 5F). However, we cannot totally rule out the possibility that Dilp2 also increases pERK levels and Dilp6 also increases pAKT levels in α′β′ MBn. Here, we propose that different intracellular signals induced by Dilp2 and Dilp6 under different feeding states, both of which suppress the hot response of α′β′ MBn (Fig 8).

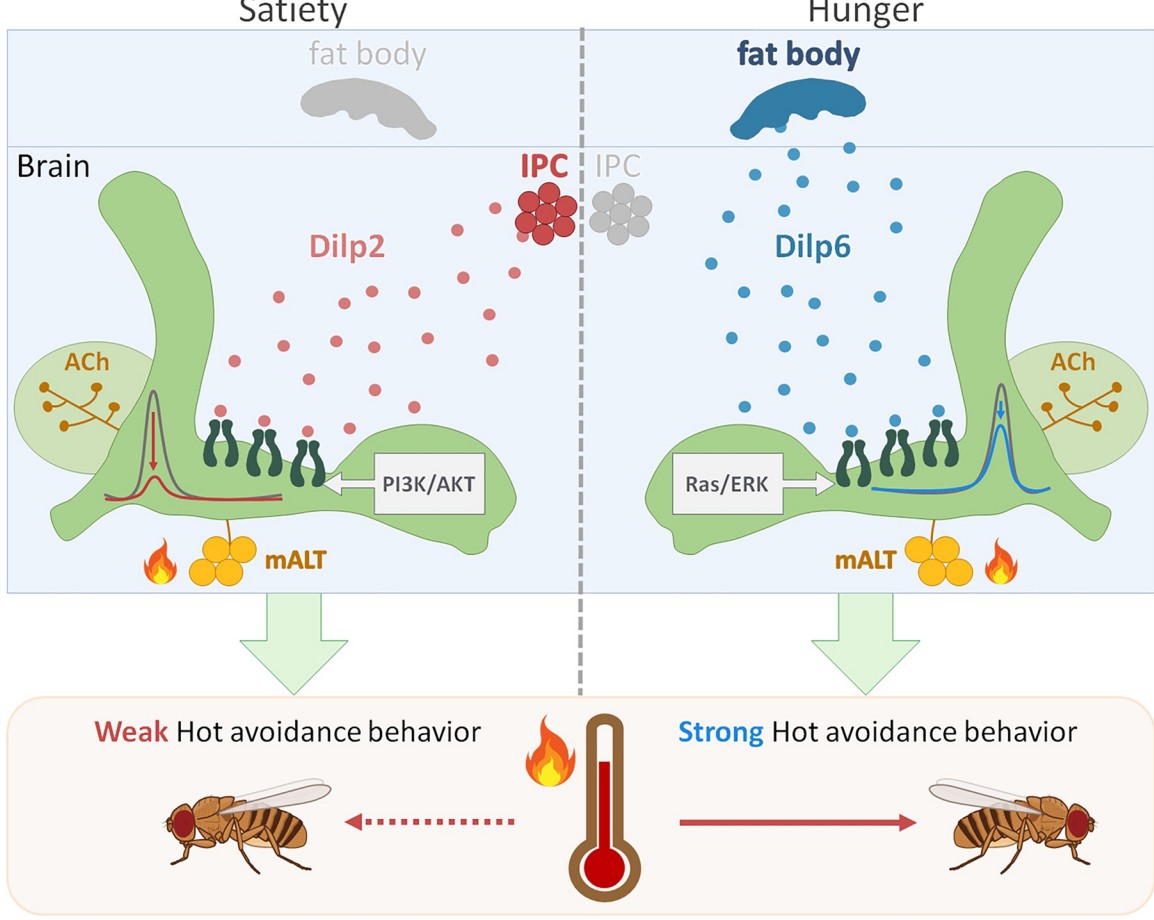

**Fig 8. The neuronal mechanism of feeding state-dependent hot avoidance in *Drosophila*.** The hot stimulus is conveyed through cholinergic mALT neurons to α′β′ MBn (green) and α′β′ MBn activity is positively correlated with HAB. In the food-sated state, Dilp2 is secreted from IPCs to inhibit the hot response of α′β′ MBn by inducing PI3K/AKT signaling that causes a weak HAB and consequently flies decrease their hot avoidance for increasing their metabolism. On the other hand, Dilp2 is no longer released from IPCs in the hungry state. Instead, Dilp6 is released from the fat body to inhibit the hot response of α′β′ MBn by inducing Ras/ERK signaling. The inhibition efficiency of Ras/ERK in the hungry state is not as strong as PI3K/AKT signaling, consequently flies exhibit strong hot responses in α′β′ MBn that contributes to a strong HAB, thereby increasing the hot avoidance of flies to reduce their metabolism for saving energy in the hungry state. HAB, hot avoidance behavior; IPC, insulin-producing cell; mALT, medial antennal lobe tract; MBn, mushroom body neuron; SEM, standard error of mean.

A previous study showed that overexpression of Dilp6 in adult fat body represses Dilp2 and Dilp5 expression in IPCs and the secretion of Dilp2 into the hemolymph [19]. Our anti-Dilp2 immunostaining data also showed that genetic knockdown of *dilp6* in the fat body increased the Dilp2 expression in IPCs. In contrast, overexpression of *dilp6* in fat body decreased Dilp2 expression in IPCs, in both sated and hungry states (S12 Fig). It has been shown that Dilp6 expression in the fat body reduces short neuropeptide F (sNPF) expression in the brain. This sNPF reduction may correlate to the Dilp2 suppression in IPCs [19,38]. Reducing circulating Dilp2 could contribute to lifespan extension and the correlated changes in carbohydrate/lipids storage and oxidative stress resistance. We postulate that Dilp6 secretion in fat body during starvation not only suppresses the Dilp2 expression in IPCs but also induces the Ras/ERK signaling in α′β′ MBn that contributes to the hungry state-dependent HAB (Fig 4).

In contrast to Dilp2, Dilp3, and Dilp5, it has been shown that Dilp1 is transiently expressed in IPCs, specifically during pupa and early stage of adult flies [39,40]. The immunohistochemistry data of Dilp1 showed that Dilp1 is expressed in newly eclosed adult flies and dramatically decreases after 1 week [39]. However, the Dilp1 expression in IPCs of adult flies lasts over 6 weeks and only slightly decreases after 7 weeks in diapause conditions [39,41]. It has been proposed that the transient Dilp1 expression during the early stage of adult flies could be associated with a metabolic transition from pupa to adult during normal conditions. Mutations of *dilp1* reduce the organism's weight during pupal development, whereas overexpression of *dilp1* increases it [40]. Survival during starvation decreases significantly in newly eclosed, but not in 6- to 7-day-old *dilp1* mutant female flies. In addition, starvation resistance is not affected in *dilp1* mutant male flies [40]. Therefore, it is unlikely that Dilp1 regulates α′β′ MBn neuronal activity for HAB since we used the 7- to 10-day-old adult flies for experiments in the current study.

A previous study showed that Dilp6 regulates AC neurons activity during starvation [18], which leads to starvation-induced reduction of $T_p$ as hungry flies prefer to stay at 23°C rather than 25°C, which is consistent with our results (Fig 1B). Manipulation of InR expression in α′β′ MBn had no effect on cold avoidance and $T_p$, suggesting that this signaling is only involved in HAB (Figs 2G, 2H, S4G and S4H). Starvation-induced Dilp6 secretion from the fat body regulates both AC neurons and α′β′ MBn, which contributes to changes in $T_p$ and HAB during the hungry state. Although 8 different Dilps are expressed in *D. melanogaster*, only single InR is present in flies, suggesting that Dilp2 and Dilp6 bind to the same InR in α′β′ MBn, but trigger distinct signaling pathways under different feeding conditions, contributing to different levels of the hot response and HAB. A prior study showed that Dilp2 and Dilp5 differentially modulate the signal transduction kinetics of pAKT in *Drosophila* S2 cells, by which Dilp2 induces acute and transient phosphorylation of AKT, whereas Dilp5 induces sustained AKT phosphorylation [42]. These results support the notion that distinct Dilp ligands could trigger different intracellular transduction kinetics through 1 InR. However, it remains unclear how Dilps binding to the same InR in α′β′ MBn induces distinct intracellular signaling under different feeding states.

Our results revealed that insulin receptors in αβ and γ MBn are not involved in HAB (S4B–S4E Fig). However, temporal silencing of the αβ or γ MBn activity reduced the HAB in the sated state (Fig 1E and 1F) suggesting that αβ and γ MBn regulate HAB via other mechanisms. Dopamine is an essential neurotransmitter produced by the central and peripheral nervous systems and plays an important role in neuromodulation [43]. Several studies have shown that dopamine is involved in thermoregulation. The dopaminergic pathways in the hypothalamus help in improving the tolerance to a high core temperature and slow down the rate of core temperature rise in the human body [44]. In rats, dopamine breakdown processes in the preoptic area and anterior hypothalamus are active and play a role in thermoregulation during

exercise [45]. In flies, the axons of 2 types of dopaminergic neurons, including the protocerebral posterior lateral (PPL) cluster and protocerebral anterior medial (PAM) cluster, innervate the MB lobes, transmitting cold signals to MBn [23,46]. It has been shown that PPL1-α3/α′3, PPL1-α2α′2, PPL1-γ2α′1, PPL1-γ1pedc, PAM-β2, and PAM-β′2 respond to cold stimuli suggesting their physiological role in conveying cold signals to MB lobes [23,46,47].

Besides temperature sensation, dopaminergic neurons also regulate the *Drosophila* feeding state and MBn activity. A previous study showed that dopaminergic neurons encode odor/taste valence and regulate internal physiology in specific MB lobe compartments [48]. Other studies have demonstrated that the dopaminergic PPL1-α′2α2 neurons only receive satiety signals, whereas PAM-β2β′2a and PPL1-γ2α′1 neurons receive both satiety and hunger signals [25,49]. Additionally, nutritious sugar feeding immediately suppresses the activity of dopaminergic PAM-γ3 neurons, providing a positive reinforcing signal for sugar-reward memory formation [50]. Together, the activity of αβ and γ MBn involved in HAB may be regulated by dopaminergic modulations during the sated state in *Drosophila*.

In *Drosophila*, hot sensation relies on thermoreceptors in the last antennal segment, the arista [4–6]. The hot signal is conveyed from the antenna lobe to the higher brain center, including the MB calyx, LH, and PLP, via hot responsive mALT projection neurons [7] (Fig 6A). Our brain imaging data showed co-localized signals in mALT axons and MB dendrites, suggesting synaptic connections between mALT and MBn (Fig 6B). In addition, immunohistochemistry data also suggest that mALT is a cholinergic neuron that transmits acetylcholine to MBn (Figs 6C and S10A). Silencing mALT neuronal activity or *ChAT* knockdown in mALT reduced HAB in sated and hungry flies, suggesting that mALT conveys hot signals in both feeding states via the cholinergic transmission (Fig 6D and 6E). Optogenetic activation of mALT via *CsChrimson* induced a significant calcium response in α′β′ MBn, whereas silencing mALT activity abolished the hot response in α′β′ MBn in both feeding states (Figs 6G and S10E). These results indicate that α′β′ MBn is downstream of mALT neurons and receives the hot input from mALT. Interestingly, our live calcium brain imaging data revealed the same hot response level in mALT neurons under both feeding states, indicating that satiety and hunger signals had no effect on the hot response in mALT (Fig 6F). Insulin signaling only suppressed the activity of α′β′ MBn, but not the upstream mALT during exposure to the hot stimulus (Figs 2–6).

Previous studies have demonstrated compartment-specific changes in synaptic strength at MBn-MBON synapses following MBn activation paired with artificial activation of reinforcing dopaminergic neurons, suggesting that the integrated information from MB is transferred to specific MBONs [34,35,51,52]. At least 9 different α′β′-related MBONs have been reported in the fly brain, and some of them play important roles in olfactory memory, visual memory, and sleep regulation in *Drosophila* [34,36,53–55]. Herein, we individually tested the involvement of each α′β′-related MBON in HAB and revealed that the activity of MBON-α′3 and MBON-β′1 is required for HAB under sated and hungry states (Fig 7A and 7B). Since the activity of α′β′ MBn was increased in response to the hot stimulus in hungry flies (Fig 1J), these flies also showed an increased calcium response to the hot stimulus in MBON-α′3 and MBON-β′1 (Fig 7C and 7D). It has been shown that MBON-α′3 forms a suppressed short-lived memory trace following aversive olfactory conditioning, whose activity is required for the 15-min memory [56]. In addition, MBON-β′1 activity is critical for polyamine odor preference and mating behaviors in female flies [57]. Our results showed the physiological roles of MBON-α′3 and MBON-β′1, which are responsive to hot stimuli and regulate HAB execution in *Drosophila* (Fig 7).

The reasons why food-sated flies prefer to stay at relatively higher temperatures compared to hungry flies remain unknown. One hypothesis is that under starvation conditions, flies try

to save their energy by changing their behaviors and becoming more sensitive to hot stimuli. In contrast, after feeding, flies become less sensitive to hot stimuli and exhibit decreased HAB to approach a relatively high temperature for increasing their metabolic rates. The detailed mechanisms of how nutritional status affects temperature-sensing behaviors in *Drosophila* remain unknown and would be an interesting topic for future studies.

## Materials and methods

### Fly stocks

$W^{1118}$ flies were used as wild-type controls. The following fly lines were used in this study: *VT30559-GAL4* (Vienna *Drosophila* Resource Center, VDRC: 206077), *VT44966-GAL4* (VDRC: 203571), *VT49246-GAL4* (Ann-Shyn Chiang's Lab), *VT30604-GAL4* (VDRC: 200228), *R16A06-GAL4* (Bloomington *Drosophila* Stock Center, BDSC: 48709), *C739-GAL4* (BDSC: 7362), *VT57244-GAL4* (VDRC: 200970), *dilp2-GAL4* (BDSC: 37516), *C305a-GAL4* (BDSC: 30829), *cg-GAL4* (BDSC: 7011), *VT40053-GAL4* (VDRC: 201352), *MB011B-GAL4* (BDSC: 68294), *MB027B-GAL4* (BDSC: 68301), *C305a-GAL4* (BDSC: 68301), *MB050B-GAL4* (BDSC: 68365), *MB051B-GAL4* (BDSC: 68275), *MB057B-GAL4* (BDSC: 68277), *MB083C-GAL4* (BDSC: 68287), *MB091C-GAL4* (Ann-Shyn Chiang's Lab), *VT0765-GAL4* (Ann-Shyn Chiang's Lab), *VT41043-GAL4* (VDRC: 200099), *MB247-lexA* (Ann-Shyn Chiang's Lab), *R35B12-lexA* (BDSC: 52735), $dilp2^{-/-}$ (BDSC: 30881), $dilp3^{-/-}$ (BDSC: 30882), $dilp5^{-/-}$ (BDSC: 30884), $dilp6^{LOF}$ (BDSC: 30885), *tublin-Gal80$^{ts}$* (Chr. 2) (BDSC: 7019), *tublin-Gal80$^{ts}$* (Chr. 3) (BDSC: 7017), *UAS-GCaMP7s* (Chr. 2) (BDSC: 80905), *UAS-GCaMP7s* (Chr. 3) (BDSC: 79032), *lexAop-GCaMP7s* (BDSC: 80913), *UAS-GtACR2* (Suewei Lin's Lab), *UAS-kir2.1::eGFP* (Suewei Lin's Lab), *UAS-mCD8:GFP* (Chr. 2) (BDSC: 5137), *UAS-mCD8:GFP* (Chr. 3) (BDSC: 5130), *UAS-CsChrimson* (BDSC: 82181), *UAS-InR$^{DN}$* (Pei-Yu Wang's Lab), *UAS-InR$^{CA}$* (Pei-Yu Wang's Lab), *UAS-dilp2$^{RNAi}$* (VDRC: 102158), *UAS-dilp2* (BDSC: 80936), *UAS-PI3K$^{DN}$* (BDSC: 8288), *UAS-AKT$^{RNAi}$* (VDRC: 103703), *UAS-AKT* (Horng-Dar Wang's Lab), *UAS-dilp6$^{RNAi}$* (VDRC: 102465), *UAS-dilp6* (Igor Vuillez's Lab), *UAS-Ras$^{RNAi}$* (VDRC: 106642), *UAS-Raf$^{RNAi}$* (VDRC: 107766), *UAS-Erk$^{RNAi}$* (VDRC: 109108), *UAS-Erk* (BDSC: 36270), and *UAS-ChAT$^{RNAi}$* (BDSC: 25856). Flies were reared on standard cornmeal food and a 12 h:12 h light:dark cycle. All flies were raised at 24˚C and 50% to 60% humidity unless stated otherwise. For inducing hunger, approximately 300 flies from desired groups were starved for 22 to 26 h with a 6 cm × 3 cm water-soaked filter paper at 24˚C. For optogenetic experiments, flies were kept in standard cornmeal food containing 400 mM all-trans-retinal for 4 days before behavioral and calcium imaging assays.

### Temperature avoidance behavior assay

The device used for analyzing temperature avoidance behavior was prepared as described previously [6,8]. Each quadrant of the device contained 4 components: a thermoelectric cooling chip with an aluminum plate (6 cm × 6 cm), a temperature sensor, a heat spreader, and a microcontroller. In first and third quadrants, the temperature was set to 25˚C, while the second and fourth quadrants were set to the experimental temperature (15 to 35˚C). The pulse-width modulation (PWM) is used in the thermoelectric device to control the operating voltage of the thermoelectric cooling module, which decides whether to turn on the power supply of cooling fans of heat spreaders according to different temperature ranges. The microcontroller sets different PWM parameters in different temperature ranges according to the characteristics of the thermoelectric cooling module. The microcontroller transmits information of the aluminum sheets' temperature values and the cooling fans' power status to a computer to monitor the real-time temperature control status. During each trial, the apparatus was covered with

glass coated with RainX to prevent flies from escaping the device, and the avoidance index was calculated for each test temperature. The assays were carried out in a room maintained at 24˚C, 50% to 60% humidity. Groups of 15 flies of 7- to 10-day-old age of both sexes were subdued with 98% $CO_2$ and randomly placed in the arena. The flies were free to move on the aluminum plate, and the movement of flies was recorded for 3 min. The avoidance index was defined as (number of flies at 25˚C—number of flies at the test temperature)/total number of flies. The avoidance indices were compared using the unpaired $t$ test (2 groups) or analysis of variance (ANOVA) (3 groups).

For adult-stage-specific gene knockdown or adult-stage-specific gene expression experiments, the *tub-GAL80^{ts}* transgene was introduced for temporal inhibition of GAL4 expression. The experimental group flies were kept at 18˚C throughout development. After eclosion, the flies were shifted to 30˚C for 5 days and shifted back to 24˚C for 12 h before behavioral assays. Our heat shock protocol did not change the HAB compared to that of the flies kept at 24˚C (S13 Fig). The control group flies were maintained at 18˚C throughout development. After eclosion, flies were shifted to 24˚C for 5.5 days before performing the behavioral assays. All behavioral assays were performed at 24˚C.

For *GtACR2*-mediated neuronal silencing experiments, temperature avoidance behaviors were recorded under blue light (468 nm) irradiation with an intensity of approximately 2.4 mW/cm² for 3 min. For *CsChrimson*-mediated neuronal activation experiments, temperature avoidance behaviors were recorded under red light (625 nm) irradiation with an intensity of approximately 11.2 mW/cm² for 3 min. Flies were received 3 min of blue or red light irradiation during the experiments.

## Immunohistochemistry and confocal imaging

Fly brains were dissected in PBS and fixed in 4% paraformaldehyde on ice with 3 repetitions of microwave irradiation (2,450 MHz; 1,100 watts) for 60 s with continuous rotation. Brain samples were then incubated in blocking buffer (PBS containing 10% normal goat serum and 2% Triton X-100) and degassed in a vacuum chamber (depressurized to 270 mmHg then held for 10 min) for 6 cycles. Next, the brains were blocked and permeabilized in blocking buffer at 25˚C for 2 h. The fly brains were immunostained with the mouse 4F3 anti-discs large (DLG) monoclonal antibody (1:10; AB-528203, Developmental Studies Hybridoma Bank, University of Iowa), mouse anti-ChAT (1:200; ChAT4B1, Developmental Studies Hybridoma Bank, University of Iowa), rabbit anti-Dilp2 (1:200; Takashi Nishimura's lab), rabbit anti-pAKT (1:200; #9271, Cell Signaling Technology), or rabbit anti-pERK (1:200; #4376, Cell Signaling Technology). Brain samples were incubated in dilution buffer (PBS containing 1% normal goat serum and 0.25% Triton X-100) with the primary antibody at 25˚C for 24 h. After 3 intensive washes in PBS-T, the samples were incubated with biotinylated goat anti-mouse IgG (1:200; B-2763, Thermo Fisher Scientific) or goat anti-rabbit IgG (1:200; B-2770, Thermo Fisher Scientific) at 25˚C for 24 h. Next, the samples were washed and incubated with Alexa Fluor 633 streptavidin (1:500; Invitrogen) at 25˚C for another 24 h. After washing, the samples were cleared and mounted using FocusClear (FC-101, CelExplorer). The samples were covered with coverslips and imaged using the Zeiss LSM 700 confocal microscope with either a 63× glycerin-immersion objective (N.A. = 1.4 and 170 μm working distance) or a 40× C-Apochromat water-immersion objective (N.A. = 1.2 and 220 μm working distance). The pinhole (optical section) was set at 1.5 μm when imaging with the 63× objective lens and at 2 μm for images taken with the 40× objective lens. All images were processed using the ZEN or ImageJ software.

## Brain calcium imaging

Live brain calcium imaging experiments were performed as described in our previous study [23]. To monitor the changes in intracellular calcium ions in response to the hot stimulus within the fly brain flies expressing GCaMP7s were immobilized using a 250 μl pipette tip. Using fine tweezers, a small opening was made on the head capsule, and a drop of adult hemolymph-like (AHL) saline (108 mM NaCl, 5 mM KCl, 2 mM $CaCl_2$, 8.2 mM $MgCl_2$, 4 mM $NaHCO_3$, 1 mM $NaH_2PO_4$, 5 mM trehalose, 10 mM sucrose, and 5 mM HEPES (pH 7.5), 265 mOsm) was added immediately to prevent dehydration of the brain. The pipette tip along with the fly was mounted in a perfusion chamber containing 400 μl AHL solution at 24˚C. The hot stimulus was administered by adding an additional 200 μl AHL solution at 55˚C. The final temperature of the solution (approximately 31˚C) was monitored using a thermometer. Time-lapse recording of the GCaMP intensity before and after the hot stimulus was performed using the Zeiss LSM700 microscope with a 40× Achroplan IR lens. The 488 nm excitation laser and a detector for emissions passing through a 555 nm short-pass filter were used for time-lapse recording. An optical slice with a resolution of $512 \times 512$ pixels was continuously monitored for 75 s at a rate of 2 frames per second. For optogenetic experiments, flies were kept in a medium containing 400 mM all-trans-retinal (R2500, SIGMA) for 4 days before performing the calcium imaging assay, as described above. A 625 nm LED was used as the light source to activate CsChrimson-expressing neurons, and GCaMP signals were recorded using the Zeiss LSM700 microscope. The regions of interest (ROIs) were manually assigned to anatomically different neuropils or soma regions and are described in detail in each figure legend. $F_0$ was defined as the average of 20 frames of fluorescence intensity before the temperature stimulus. ΔF was defined as fluorescence intensity changes after temperature stimulus, which is fluorescence intensity post stimulation minus $F_0$. The temperature-induced intensity changes were calculated as $\Delta F/F_0$, and intensity maps were generated using the ImageJ software; the maximum (max) intensity (8-bit binary digit, 255) is presented by red color, and minimum intensity (8-bit binary digit, 0) is presented by blue color.

## Statistical analysis

Raw data were analyzed parametrically using the Prism 6.0 software (GraphPad). Raw data from 2 groups were evaluated using the unpaired two-tailed *t* test. Raw data from 3 groups were evaluated using one-way analysis of variance (ANOVA) and Tukey's multiple comparison tests. A *P*-value <0.05 was considered statistically significant. The *N* values for each experiment are indicated in the figures. All data are presented as mean ± standard error of mean (SEM).

## Supporting information

**S1 Fig. The introduction of thermoelectric device.** (**A**, **B**) The top view (**A**) and side view (**B**) of a thermoelectric device. The thermoelectric device contains 6 components: microcontroller system, temperature sensors, aluminum plates, cooling chips, heat spreaders, and a glass cover. The pulse-width modulation (PWM) is used to control the operating voltage of thermoelectric cooling module. The thermoelectric cooling module is used to control the temperature of aluminum plate that range from 15˚C to 35˚C. We set a 6 × 6 cm aluminum sheet on each thermoelectric cooling module to increase conduction velocity. For heat dissipation, we set a heat spreader with aluminum below each cooling chip. Four thermoelectric cooling modules are lined up to form an arena and temperature sensor is added in each thermoelectric cooling module. (**C**) The distributions of flies on the thermoelectric device during a two-choice assay experiment. (**D**) The accuracy and stability of the thermoelectric device has been shown. The

2' (light-red region) and 4' (dark-red region) quadrants were individually set to different test temperatures (15, 17, 19, 21, 23, 27, 29, 31, 33, or 35°C). The 1' (dark-green region) and 3' (light-green region) quadrants were set to 25°C. The temperatures on each of the 4 quadrants plates were recorded for 3,600 s. The margin of error on each aluminum plate was lower than 1°C (approximately 0.5°C) in all test temperature settings. (**E**, **F**) Three min of blue or red light irradiations did not alter the setting temperatures of the aluminum plates of the thermoelectric device. The 2' (light-red region) and 4' (dark-red region) quadrants were set to different test temperatures (27, 29, 31, or 33°C). The 1' (dark-green region) and 3' (light-green region) quadrants were set to 25°C. The temperature was recorded for 540 s on each aluminum plate, and light irradiations were added from 180 to 360 s. The margin of error on each aluminum plate was lower than 1°C in blue light (**E**) and red light (**F**) irradiation conditions. The data underlying this figure can be found in S1 Data.
(TIF)

**S2 Fig. MBn activity is critical for HAB.** (**A**) Silencing MBn activity via *VT30559-GAL4 > UAS-kir2.1* reduced HAB during both sated and hungry states (Satiety: *P*-values: 0.033, <0.0001, 0.0073, and 0.1177 from left to right; Hunger: *P*-values: 0.0002, 0.0097, 0.0002, and 0.4309 from left to right). (**B**) Hungry flies showed significantly higher HAB than sated flies under blue light irradiation (*P*-values: 0.0832, 0.3191, 0.2438, 0.202, 0.0006, 0.7372, 0.0003, 0.0124, 0.0058, 0.0387, and 0.0569 from left to right). (**C**) Hungry flies showed significantly higher HAB than sated flies under red light irradiation. (*P*-values: 0.7901, 0.9453, 0.8403, 0.6582, 0.0015, 0.4891, 0.002, 0.0231, 0.0004, 0.0101, and 0.0962 from left to right). (**D**) Silencing γ MBn activity via *R16A06-GAL4 > UAS-GtACR2* reduced HAB during the sated state (Satiety: *P*-values: <0.0001, 0.0005, 0.0024, and 0.1123 from left to right; Hunger: *P*-values: 0.5773, 0.7602, 0.1986, and 0.9307 from left to right). (**E**) Silencing αβ MBn activity via *C739-GAL4 > UAS-GtACR2* reduced HAB during the sated state (Satiety: *P*-values: 0.0002, 0.0001, 0.0002, and < 0.0001 from left to right; Hunger: *P*-values: 0.8762, 0.7292, 0.684, and 0.422 from left to right). (**F**) Silencing α′β′ MBn activity via *VT57244-GAL4 > UAS-GtACR2* reduced HAB during sated and hungry states (Satiety: *P*-values: <0.0001, 0.0009, <0.0001, and <0.0001 from left to right; Hunger: *P*-values: 0.0009, 0.0111, <0.0001, and 0.0001 from left to right). (**G**) α′β′ MBn activation via *VT30604-GAL4 > UAS-CsChrimson* increased HAB (Satiety: *P*-values: <0.0001, 0.0058, 0.0005, and 0.1063 from left to right; Hunger: *P*-values: 0.0082, 0.0116, 0.0011, and 0.4237 from left to right). (**H**) Room temperature stimuli did not induce a significant calcium response in γ, αβ, and α′β′ MBn. The GCaMP intensity changes ($\Delta F/F_0$) in γ, αβ, and α′β′ MBn horizontal lobes were recorded and analyzed. There were no significant differences in GCaMP intensity in γ, αβ, and α′β′ MBn before and after room temperature stimuli (*P*-values: 0.9902, 0.298, and 0.2444 from left to right). The arrow under each calcium response curve indicates the time points at which the room temperature stimuli (24°C) were applied. Each *N* represents either a group of 15 flies analyzed together in behavioral assays (**A**–**G**) or a single fly in calcium imaging experiments (**H**). Data are represented as mean ± SEM with dots representing individual values. The data underlying this figure can be found in S1 Data. Data were analyzed by one-way ANOVA followed by Tukey's test (**A**, **D**, **E**–**G**) or unpaired two-tailed *t* test (**B**, **C**, **H**), *$P < 0.05$.
(TIF)

**S3 Fig. GFP expression patterns in flies with different GAL4 drivers.** (**A**) MBn with *VT30559-GAL4* expression. (**B**) γ MBn with *VT44966-GAL4* expression. (**C**) αβ MBn with *VT49246-GAL4* expression. (**D**) α′β′ MBn with *VT30604-GAL4* expression. (**E**) γ MBn with *R16A06-GAL4* expression. (**F**) αβ MBn with *C739-GAL4* expression. (**G**) α′β′ MBn with *VT57244-GAL4* expression. (**H**) MBON-γ5β′2a with *MB011B-GAL4* expression. (**I**) MBON-

α′3 with *MB027B-GAL4* expression. (**J**) MBON-α′1 with *MB050B-GAL4* expression. (**K**) MBON-γ2α′1 with *MB051B-GAL4* expression. (**L**) MBON-β′1 with *MB057B-GAL4* expression. (**M**) MBON-γ3β′1 with *MB083C-GAL4* expression. (**N**) MBON-α′2 with *MB091C-GAL4* expression. (**O**) MBON-β2β′2a with *VT0765-GAL4* expression. (**P**) MBON-β′2 with *VT41043-GAL4* expression. Each GAL4 line was crossed with the *UAS-mCD8::GFP; UAS-mCD8*::*GFP* reporter line and confocal brain imaging was performed on the progeny. Brain neuropils were counterstained with anti-DLG antibody (magenta). Scale bar, 50 μm.
(TIF)

**S4 Fig. InR manipulations in α′β′ MBn affects HAB.** (**A**) Genetic expression of *InR^{DN}* in α′β′ MBn via *VT57244-GAL4 > UAS-InR^{DN}* increased HAB during both sated and hungry states (Satiety: *P*-values: 0.0003, <0.0001, <0.0001, and 0.0003 from left to right; Hunger: *P*-values: <0.0001, <0.0001, <0.0001, and 0.0002 from left to right). (**B**) Genetic expression of *InR^{DN}* in γ MBn via *VT44966-GAL4 > UAS-InR^{DN}* had no effect on HAB during both sated and hungry states (Satiety: *P*-values: 0.3221, 0.9858, 0.6006, and 0.8822 from left to right; Hunger: *P*-values: 0.9897, 0.6707, 0.8446, and 0.5868 from left to right). (**C**) Genetic expression of *InR^{DN}* in γ MBn via *R16A06-GAL4 > UAS-InR^{DN}* had no effect on HAB during both sated and hungry states (Satiety: *P*-values: 0.3765, 0.8375, 0.9045, and 0.9183 from left to right; Hunger: *P*-values: 0.5816, 0.8305, 0.8359, and 0.7554 from left to right). (**D**) Genetic expression of *InR^{DN}* in αβ MBn via *VT49246-GAL4 > UAS-InR^{DN}* had no effect on HAB during both sated and hungry states (Satiety: *P*-values: 0.9815, 0.4656, 0.9837, and 0.9286 from left to right; Hunger: *P*-values: 0.9478, 0.9479, 0.9743, and 0.9928 from left to right). (**E**) Genetic expression of *InR^{DN}* in αβ MBn via *C739-GAL4 > UAS-InR^{DN}* had no effect on HAB during both sated and hungry states (Satiety: *P*-values: 0.9846, 0.7737, 0.7182, and 0.7518 from left to right; Hunger: *P*-values: 0.5773, 0.6921, 0.853, and 0.4112 from left to right). (**F**) Permissive temperature control of Fig 2E. At permissive temperatures, there were no significant differences in HAB between *tub-GAL80^{ts}; VT30604-GAL4 > InR^{DN}*, *tub-GAL80^{ts}; VT30604-GAL4 >+*, and *UAS-InR^{DN} >+* flies during sated and hungry states (Satiety: *P*-values: 0.4415, 0.7193, 0.7097, and 0.9129 from left to right; Hunger: *P*-values: 0.9994, 0.7927, 0.9944, and 0.3847 from left to right). (**G**) Genetic expression of *InR^{DN}* in α′β′ MBn via *VT30604-GAL4 > UAS-InR^{DN}* had no effect on the cold avoidance behavior during sated and hungry states (Satiety: *P*-values: 0.4082, 0.9822, 0.9626, and 0.7241 from left to right; Hunger: *P*-values: 0.5038, 0.9253, 0.7114, and 0.6802 from left to right). (**H**) Genetic expression of *InR^{DN}* in α′β′ MBn via *VT57244-GAL4 > UAS-InR^{DN}* had no effect on the cold avoidance behavior during both sated and hungry states (Satiety: *P*-values: 0.6194, 0.7396, 0.8875, and 0.8173 from left to right; Hunger: *P*-values: 0.5038, 0.5851, 0.6177, and 0.7691 from left to right). Each *N* represents a group of 15 flies analyzed together in the behavioral assay. The data underlying this figure can be found in S1 Data. Data are represented as mean ± SEM with dots representing individual values and analyzed by one-way ANOVA followed by Tukey's test, *\*P* < 0.05.
(TIF)

**S5 Fig. Additional experiments for InR manipulations in α′β′ MBn.** (**A**) Genetic expression of *InR^{CA}* in α′β′ MBn via *VT57244-GAL4 > UAS-InR^{CA}* decreased HAB during both sated and hungry states (Satiety: *P*-values: 0.0006, <0.0001, <0.0001, and <0.0001 from left to right; Hunger: *P*-values: 0.0002, <0.0001, 0.0039, and 0.8281 from left to right). (**B**) Permissive temperature control of Fig 2F. At permissive temperatures, there were no significant differences in HAB between *tub-GAL80^{ts}; VT30604-GAL4 > InR^{CA}*, *tub-GAL80^{ts}; VT30604-GAL4 >+*, and *UAS-InR^{CA} >+* flies during both sated and hungry states (Satiety: *P*-values: 0.919, 0.503, 0.5766, and 0.6841 from left to right; Hunger: *P*-values: 0.2961, 0.8591, 0.7777, and 0.9495 from left to right). (**C**) Genetic expression of *InR^{CA}* in α′β′ MBn did not affect cold avoidance

behavior in both satiety and hungry states (Satiety: *P*-values: 0.9561, 0.8529, 0.2998, and 0.655 from left to right; Hunger: *P*-values: 0.7102, 0.9054, 0.843, and 0.2119 from left to right). (**D**) The morphology of α′β′ MBn, labeled by GFP (green), was not affected in flies carrying *UAS-mCD8::GFP/UAS-InR<sup>DN</sup>; VT30604-GAL4/+* or *UAS-mCD8::GFP/UAS-InR<sup>CA</sup>; VT30604-GAL4/+*. Each *N* represents a group of 15 flies analyzed together in the behavioral assay (**A**–**C**). The data underlying this figure can be found in S1 Data. Data are represented as mean ± SEM with dots representing individual values and analyzed by one-way ANOVA followed by Tukey's test, \*$P < 0.05$.
(TIF)

**S6 Fig. Manipulation of *dilp2* in IPCs affects HAB in sated state.** (**A**) HAB was not affected in *dilp3* mutant flies (*dilp3<sup>-/-</sup>*) and *dilp5* mutant flies (*dilp5<sup>-/-</sup>*) (Satiety: *P*-values: 0.6672, 0.4866, 0.8345, and 0.4314 from left to right; Hunger: *P*-values: 0.1133, 0.2915, 0.718, and 0.426 from left to right). (**B**) *dilp2-GAL4 > UAS-dilp2<sup>RNAi</sup>* flies exhibited an increased HAB during the sated state (Satiety: *P*-values: <0.0001, <0.0001, <0.0001, and <0.0001 from left to right; Hunger: *P*-values: 0.6612, 0.7088, 0.2353, and 0.8943 from left to right). (**C**) Permissive temperature control of **Fig 3C**. At permissive temperatures, there were no significant differences in HAB between *dilp2-GAL4; tub-GAL80<sup>ts</sup> > UAS-dilp2<sup>RNAi</sup>*, *dilp2-GAL4; tub-GAL80<sup>ts</sup> > +*, and *UAS-dilp2<sup>RNAi</sup> > +* flies during the sated state (Satiety: *P*-values: 0.8918, 0.7176, 0.5073, and 0.7762 from left to right). (**D**) Immunostaining with anti-Dilp2 antibody in *dilp2-GAL4 > UAS-mCD8::GFP* flies (left panel). Quantification of anti-Dilp2 antibody immunostaining intensity in GFP positive IPCs during sated and hungry states (right panel). The anti-Dilp2 immunostaining signals in IPCs were normalized to the signals in fan-shaped body ($P < 0.0001$). Scale bar, 20 μm. (**E**) Hot stimulus did not induce the calcium response in IPCs during both feeding states. The soma of IPCs were recorded and analyzed. There were no significant differences in GCaMP intensity in IPCs before and after hot stimuli (*P*-values: 0.8381). The arrow under calcium response curve indicates the time points at which the hot stimuli were applied. (**F**) Permissive temperature control of **Fig 3E**. At permissive temperatures, there were no significant differences in HAB between *dilp2-GAL4; tub-GAL80<sup>ts</sup> > UAS-dilp2*, *dilp2-GAL4; tub-GAL80<sup>ts</sup> > +*, and *UAS-dilp2 > +* flies during sated and hungry states (Satiety: *P*-values: 0.5363, 0.3002, 0.8416, and 0.7235 from left to right; Hunger: *P*-values: 0.3413, 0.8279, 0.9225, and 0.4409 from left to right). Each *N* represents either a group of 15 flies analyzed together in the behavioral assay (**A**, **B**, **C**, **F**) or a single fly in Dilp2 immunostaining experiments (**D**) and calcium imaging experiments (**E**). The data underlying this figure can be found in S1 Data. Data are represented as mean ± SEM with dots representing individual values and analyzed by one-way ANOVA followed by Tukey's test (**A**, **B**, **C**, **F**) or unpaired two-tailed *t* test (**D**, **E**). \*$P < 0.05$; ns, not significant.
(TIF)

**S7 Fig. Manipulation of PI3K/AKT signaling in α′β′ MBn affects HAB in sated state.** (**A**) Genetic expression of *PI3K<sup>DN</sup>* in α′β′ MBn via *VT30604-GAL4 > UAS-PI3K<sup>DN</sup>* increased HAB during the sated state (Satiety: *P*-values: 0.0055, 0.0072, 0.0341, and 0.2073 from left to right; Hunger: *P*-values: 0.6698, 0.7836, 0.6583, and 0.5004 from left to right). (**B**) Genetic expression of *PI3K<sup>DN</sup>* in α′β′ MBn via *VT57244-GAL4 > UAS-PI3K<sup>DN</sup>* increased HAB during the sated state (Satiety: *P*-values: 0.0002, <0.0001, <0.0001, and 0.002 from left to right; Hunger: *P*-values: 0.4156, 0.3309, 0.6178, and 0.5198 from left to right). (**C**) RNAi-mediated knockdown of *AKT* in α′β′ MBn via *VT30604-GAL4 > UAS-AKT<sup>RNAi</sup>* increased HAB during the sated state (Satiety: *P*-values: <0.0001, <0.0001, 0.002, and 0.0018 from left to right; Hunger: *P*-values: 0.9554, 0.6714, 0.9494, and 0.1951 from left to right). (**D**) RNAi-mediated knockdown of *AKT* in α′β′ MBn via *VT57244-GAL4 > UAS-AKT<sup>RNAi</sup>* increased HAB during the sated state

(Satiety: *P*-values: <0.0001, <0.0001, <0.0001, and <0.0001 from left to right; Hunger: *P*-values: 0.6866, 0.3042, 0.5841, and 0.588 from left to right). (**E**) Adult-stage-specific expression of *PI3K*$^{DN}$ in α′β′ MBn via *tub-GAL80*$^{ts}$*; VT30604-GAL4 > UAS-PI3K*$^{DN}$ increased HAB during the sated state (Satiety: *P*-values: <0.0001, <0.0001, 0.0003, and 0.2743 from left to right; Hunger: *P*-values: 0.9187, 0.9098, 0.268, and 0.1859 from left to right). (**F**) At permissive temperatures, there were no significant differences in HAB between *tub-GAL80*$^{ts}$*; VT30604-GAL4 > UAS-PI3K*$^{DN}$, *tub-GAL80*$^{ts}$*; VT30604-GAL4 > +*, and *UAS-PI3K*$^{DN}$ *> +* flies during the sated state (Satiety: *P*-values: 0.9063, 0.7536, 0.7213, and 0.5989 from left to right). (**G**) Permissive temperature control of Fig 3G. At permissive temperatures, there were no significant differences in HAB between *tub-GAL80*$^{ts}$*; VT30604-GAL4 > UAS-AKT*$^{RNAi}$, *tub-GAL80*$^{ts}$*; VT30604-GAL4 > +*, and *UAS-AKT*$^{RNAi}$ *> +* flies during the sated state (Satiety: *P*-values: 0.5946, 0.1219, 0.4254, and 0.3847 from left to right). (**H**) Genetic expression of *AKT* in α′β′ MBn via *VT30604-GAL4 > UAS-AKT* decreased HAB during both sated and hungry states (Satiety: *P*-values: <0.0001, 0.0054, 0.0006, and 0.0918 from left to right; Hunger: *P*-values: <0.0001, 0.0005, 0.0001, and 0.0588 from left to right). (**I**) Permissive temperature control of Fig 3I. At permissive temperatures, there were no significant differences in HAB between *tub-GAL80*$^{ts}$*; VT30604-GAL4 > UAS-AKT*, *tub-GAL80*$^{ts}$*; VT30604-GAL4 > +* and *UAS-AKT > +* flies during both sated and hungry states (Satiety: *P*-values: 0.3167, 0.2275, 0.9931, and 0.7261 from left to right; Hunger: *P*-values: 0.6118, 0.8815, 0.7386, and 0.3188 from left to right). Each *N* represents a group of 15 flies analyzed in the behavioral assay. The data underlying this figure can be found in S1 Data. Data are represented as mean ± SEM with dots representing individual values. Data were analyzed by one-way ANOVA followed by Tukey's test. \**P* < 0.05.
(TIF)

**S8 Fig. Manipulation of *dilp6* in the fat body affects HAB in hungry state.** (**A**) Genetic knockdown of *dilp6* in the fat body via *cg-GAL4 > UAS-dilp6*$^{RNAi}$ increased HAB during the hungry state (Satiety: *P*-values: 0.746, 0.8274, 0.8812, and 0.5568 from left to right; Hunger: *P*-values: <0.0001, 0.0007, <0.0001, and 0.0338 from left to right). (**B**) Permissive temperature control of Fig 4C. At permissive temperatures, there were no significant differences in HAB between *cg-GAL4; tub-GAL80*$^{ts}$ *> UAS-dilp6*$^{RNAi}$, *cg-GAL4; tub-GAL80*$^{ts}$ *>+*, and *UAS-dilp6*$^{RNAi}$ *> +* flies during the hungry state (Hunger: *P*-values: 0.9535, 0.8192, 0.305, and 0.576 from left to right). (**C**) Permissive temperature control of Fig 4E. At permissive temperatures, there were no significant differences in HAB between *cg-GAL4; tub-GAL80*$^{ts}$ *> UAS-dilp6*, *cg-GAL4; tub-GAL80*$^{ts}$ *>+*, and *UAS-dilp6 > +* flies during the hungry state (Hunger: *P*-values: 0.3979, 0.6643, 0.475, and 0.3847 from left to right). Each *N* represents a group of 15 flies analyzed together in the behavioral assay. Data are represented as mean ± SEM with dots representing individual values. The data underlying this figure can be found in S1 Data. Data were analyzed by one-way ANOVA followed by Tukey's test. \**P* < 0.05.
(TIF)

**S9 Fig. Additional experiments for Ras/ERK manipulations in α′β′ MBn.** (**A**) Genetic knockdown of *Ras* in α′β′ MBn via *VT30604-GAL4 > UAS-Ras*$^{RNAi}$ increased HAB during the hungry state (Satiety: *P*-values: 0.1516, 0.2869, 0.4018, and 0.8412 from left to right; Hunger: *P*-values: <0.0001, <0.0001, 0.0107, and 0.0599 from left to right). (**B**) Genetic knockdown of *Ras* in α′β′ MBn via *VT57244-GAL4 > UAS-Ras*$^{RNAi}$ increased HAB during the hungry state (Satiety: *P*-values: 0.7096, 0.7749, 0.7044, and 0.8978 from left to right; Hunger: *P*-values: <0.0001, 0.0002, 0.0006, and 0.0031 from left to right). (**C**) Genetic knockdown of *Raf* in α′β′ MBn via *VT30604-GAL4 > UAS-Raf*$^{RNAi}$ increased HAB during the hungry state (Satiety: *P*-values: 0.7734, 0.2772, 0.197, and 0.6603 from left to right; Hunger: *P*-values: 0.0002, <0.0001,

0.0001, and 0.0356 from left to right). (**D**) Genetic knockdown of *Raf* in α′β′ MBn via *VT57244-GAL4 > UAS-Raf^RNAi^* increased HAB during the hungry state (Satiety: *P*-values: 0.4945, 0.7428, 0.5287, and 0.5035 from left to right; Hunger: *P*-values: 0.0099, <0.0001, <0.0001, and 0.0111 from left to right). (**E**) Genetic knockdown of *Erk* in α′β′ MBn via *VT30604-GAL4 > UAS-Erk^RNAi^* increased HAB during the hungry state (Satiety: *P*-values: 0.3408, 0.8722, 0.3893, and 0.8622 from left to right; Hunger: *P*-values: <0.0001, 0.0003, 0.0298, and 0.0081 from left to right). (**F**) Genetic knockdown of *Erk* in α′β′ MBn via *VT57244-GAL4 > UAS-Erk^RNAi^* increased HAB during the hungry state (Satiety: *P*-values: 0.845, 0.9818, 0.9571, and 0.5833 from left to right; Hunger: *P*-values: 0.0133, <0.0001, 0.0064, and 0.0386 from left to right). (**G**) Adult-stage-specific knockdown of *Ras* in α′β′ MBn via *tub-GAL80^ts^; VT30604-GAL4 > UAS-Ras^RNAi^* increased HAB during the hungry state (Satiety: *P*-values: 0.4816, 0.8399, 0.8754, and 0.2618 from left to right; Hunger: *P*-values: <0.0001, 0.0058, 0.0011, and 0.0091 from left to right). (**H**) At permissive temperatures, there were no significant differences in HAB between *tub-GAL80^ts^; VT30604-GAL4 > UAS-Ras^RNAi^*, *tub-GAL80^ts^; VT30604-GAL4 > +*, and *UAS-Ras^RNAi^ > +* flies during the hungry state (Hunger: *P*-values: 0.4564, 0.6483, 0.7827, and 0.9778 from left to right). (**I**) Adult-stage-specific knockdown of *Raf* in α′β′ MBn via *tub-GAL80^ts^; VT30604-GAL4 > UAS-Raf^RNAi^* increased HAB during the hungry state (Satiety: *P*-values: 0.2939, 0.1489, 0.3362, and 0.0771 from left to right; Hunger: *P*-values: <0.0001, 0.0003, 0.0243, and 0.1183 from left to right). (**J**) At permissive temperatures, there were no significant differences in HAB between *tub-GAL80^ts^; VT30604-GAL4 > UAS-Raf^RNAi^*, *tub-GAL80^ts^; VT30604-GAL4 > +*, and *UAS-Raf^RNAi^ > +* flies during the hungry state (Hunger: *P*-values: 0.4823, 0.6508, 0.6992, and 0.4289 from left to right). (**K**) Permissive temperature control of **Fig 4G**. At permissive temperatures, there were no significant differences in HAB between *tub-GAL80^ts^; VT30604-GAL4 > UAS-Erk^RNAi^*, *tub-GAL80^ts^; VT30604-GAL4 > +*, and *UAS-Erk^RNAi^ > +* flies during the hungry state (Hunger: *P*-values: 0.3866, 0.8864, 0.5685, and 0.6205 from left to right). (**L**) Genetic expression of *Erk* in α′β′ MBn via *VT30604-GAL4 > UAS-Erk* decreased HAB during the hungry state (Satiety: *P*-values: 0.5993, 0.617, 0.5746, and 0.7076 from left to right; Hunger: *P*-values: <0.0001, 0.0005, 0.0002, and 0.0561 from left to right). (**M**) Permissive temperature control of **Fig 4I**. At permissive temperatures, there were no significant differences in HAB between *tub-GAL80^ts^; VT30604-GAL4 > UAS-Erk*, *tub-GAL80^ts^; VT30604-GAL4 > +*, and *UAS-Erk > +* flies during the hungry state (Hunger: *P*-values: 0.7216, 0.6452, 0.1392, and 0.4289 from left to right). Each *N* represents a group of 15 flies analyzed together in the behavioral assay. Data are represented as mean ± SEM with dots representing individual values. The data underlying this figure can be found in S1 Data. Data were analyzed by one-way ANOVA followed by Tukey's test. *\*P* < 0.05.
(TIF)

**S10 Fig. Blocking neuronal activity or silencing *ChAT* in mALT reduces HAB.** (**A**) Immunostaining with ChAT antibody in *UAS-mCD8::GFP; VT40053-GAL4 > +* (*VT40053 > +*) and *UAS-mCD8::GFP; VT40053-GAL4 > UAS-ChAT^RNAi^* (*VT40053 > ChAT^RNAi^*) flies (left panel). Quantification of anti-ChAT antibody immunostaining intensity in GFP positive signals of the calyx (right panel). The anti-ChAT immunostaining signals in the calyx were normalized to the signals in the peduncle (*P* < 0.0001). Scale bar, 10 μm. (**B**) Silencing mALT neuronal activity via Kir2.1 expression (*VT40053-GAL4 > UAS-kir2.1*) decreased HAB during both sated and hungry states (Satiety: *P*-values: 0.0063, 0.0008, <0.0001, and <0.0001 from left to right; Hunger: *P*-values: <0.0001, <0.0001, <0.0001, and <0.0001 from left to right). (**C**) Genetic knockdown of *ChAT* in mALT neurons (*VT40053-GAL4 > UAS-ChAT^RNAi^*) decreased HAB during both sated and hungry states (Satiety: *P*-values: <0.0001, 0.0015, 0.0059, and <0.0001

from left to right; Hunger: *P*-values: 0.0091, <0.0001, 0.0037, and 0.0222 from left to right). (**D**) Permissive temperature control of [Fig 6E]. At permissive temperatures, there were no significant differences in HAB between *tub-GAL80ts; VT40053-GAL4 > UAS-ChAT^RNAi*, *tub-GAL80ts; VT40053-GAL4 > +* and *UAS-ChAT^RNAi > +* flies during both feeding states (Satiety: *P*-values: 0.0676, 0.4831, 0.524, and 1 from left to right; Hunger: *P*-values: 0.1797, 0.9068, 0.1219, and 0.3847 from left to right). (**E**) Hot stimulus induced calcium responses in α′β′ MBn (black), while silencing mALT activity via Kir2.1 expression inhibited hot-induced calcium responses in α′β′ MBn (red) (*P* = 0.0002 for satiety; *P* = 0.0101 for hunger). The GCaMP intensity changes ($\Delta F/F_0$) in MB β′ lobe were recorded and analyzed. The arrows under each calcium response curve indicate the time points at which the hot stimulus was applied. Each *N* represents a single fly in ChAT immunostaining experiments (**A**) and calcium imaging experiments (**E**) or group of 15 flies analyzed together in the behavioral assay (**B–D**). Data are represented as mean ± SEM with dots representing individual values. The data underlying this figure can be found in [S1 Data]. Data were analyzed by the unpaired two-tailed *t* test (**A, E**) or one-way ANOVA followed by Tukey's test (**B–D**). *$P < 0.05$; ns, not significant. (TIF)

**S11 Fig. Identification of MBONs that are required for HAB execution.** (**A**) Optogenetic silencing of MBON-γ5β′2a activity via *GtACR2* expression (*MB011B-GAL4 > UAS-GtACR2*) had no effect on HAB during both sated and hungry states (Satiety: *P*-values: 0.7164, 0.971, 0.6755, and 0.9396 from left to right; Hunger: *P*-values: 0.9746, 0.9884, 0.9782, and 0.867 from left to right). (**B**) Optogenetic silencing of MBON-α′1 activity via *GtACR2* expression (*MB050B-GAL4 > UAS-GtACR2*) had no effect on HAB during both sated and hungry states (Satiety: *P*-values: 0.6302, 0.6589, 0.4405, and 0.7501 from left to right; Hunger: *P*-values: 0.6951, 0.5973, 0.9209, and 0.7727 from left to right). (**C**) Optogenetic silencing of MBON-γ2α′1 activity via *GtACR2* expression (*MB051B-GAL4 > UAS-GtACR2*) had no effect on HAB during both sated and hungry states (Satiety: *P*-values: 0.4358, 0.6369, 0.2102, and 0.5801, from left to right; Hunger: *P*-values: 0.7697, 0.903, 0.8262, and 0.9061, from left to right). (**D**) Optogenetic silencing of MBON-γ3β′1 activity via *GtACR2* expression (*MB083C-GAL4 > UAS-GtACR2*) had no effect on HAB during both sated and hungry states (Satiety: *P*-values: 0.657, 0.2801, 0.4716, and 0.1395, from left to right; Hunger: *P*-values: 0.9009, 0.6699, 0.7454, and 0.9781 from left to right). (**E**) Optogenetic silencing of MBON-α′2 activity via *GtACR2* expression (*MB091C-GAL4 > UAS-GtACR2*) had no effect on HAB during both sated and hungry states (Satiety: *P*-values: 0.4278, 0.7957, 0.8411, and 0.6193 from left to right; Hunger: *P*-values: 0.0823, 0.5552, 0.1326, and 0.602, from left to right). (**F**) Optogenetic silencing of MBON-β2β′2a activity via *GtACR2* expression (*VT0765-GAL4 > UAS-GtACR2*) had no effect on HAB during both sated and hungry states (Satiety: *P*-values: 0.8509, 0.8318, 0.9641, and 0.1453 from left to right; Hunger: *P*-values: 0.9756, 0.7681, 0.2891, and 0.5207, from left to right). (**G**) Optogenetic silencing of MBON-β′2 activity via *GtACR2* expression (*VT41043-GAL4 > UAS-GtACR2*) had no effect on HAB during both sated and hungry states (Satiety: *P*-values: 0.759, 0.684, 0.1537, and 0.3021, from left to right; Hunger: *P*-values: 0.6198, 0.4244, 0.4526, and 0.7793, from left to right). Each *N* represents a group of 15 flies analyzed together in the behavioral assay. The data underlying this figure can be found in [S1 Data]. Data are represented as mean ± SEM with dots representing individual values and analyzed by one-way ANOVA. (TIF)

**S12 Fig. Manipulation of *dilp6* in the fat body affects Dilp2 expression in IPCs.** Immunostaining with anti-Dilp2 antibody in *cg-GAL4 > UAS-dilp6^RNAi* (*dilp6^RNAi*), *cg-GAL4 > +* (Control), and *cg-GAL4 > UAS-dilp6* (*dilp6*) flies (left panel). Quantification of anti-Dilp2

immunopositive signals in IPC during sated and hungry states (right panel). The anti-Dilp2 immunostaining signals in IPCs were normalized to the signals in the fan-shaped body (Satiety: $P$-values: <0.0001 and 0.0077 from left to right; Hunger: $P$-values: <0.0001 and 0.0002 from left to right). Scale bar, 20 μm. Each $N$ represents a single fly in Dilp2 immunostaining experiments. Data are represented as mean ± SEM with dots representing individual values. The data underlying this figure can be found in S1 Data. Data were analyzed by unpaired two-tailed $t$ test. $^*P < 0.05$.
(TIF)

**S13 Fig. Normal HAB after chronic temperature shifts.** Wild-type flies were raised under a constant temperature of either 24°C (non-heat shock) or 18°C during embryonic and larval development, transferred to 30°C for 5 days after adult eclosion, and then shifted back to 24°C for 12 h before the HAB assay was conducted (heat shock). There were no significant differences in HAB between non-heat shock and heat shock groups in both feeding states (Satiety: $P$-values: 0.9241, 0.8008, 0.7063, and 0.5931 from left to right; Hunger: $P$-values: 0.3508, 0.7399, 0.2583, and 0.6540 from left to right). Each $N$ represents a group of 15 flies analyzed together in the behavioral assay. Data are represented as mean ± SEM with dots representing individual values. The data underlying this figure can be found in S1 Data. Data were analyzed by unpaired two-tailed $t$ test.
(TIF)

**S1 Video. Hot avoidance behavior in fruit flies.**
(MP4)

**S1 Data. Source data for graph in this paper.**
(XLSX)

## Acknowledgments

We thank the Bloomington *Drosophila* Stock Center (BDSC), Vienna *Drosophila* RNAi Center, Vienna Tile (VT) Library, Fly Core in Taiwan, Suewei Lin, Pei-Yu Wang, Horng-Dar Wang, Ann-Shyn Chiang, and Igor Vuillez for providing the fly stocks. We also thank Takashi Nishimura for providing us with the anti-Dilp2 antibody.

## Author Contributions

**Conceptualization:** Meng-Hsuan Chiang, Yu-Chun Lin, Chia-Lin Wu.

**Data curation:** Meng-Hsuan Chiang, Yu-Chun Lin, Peng-Shiuan Lee.

**Formal analysis:** Meng-Hsuan Chiang, Yu-Chun Lin.

**Funding acquisition:** Chia-Lin Wu.

**Investigation:** Meng-Hsuan Chiang, Yu-Chun Lin, Chia-Lin Wu.

**Methodology:** Meng-Hsuan Chiang, Yu-Chun Lin, Sheng-Fu Chen, Tsai-Feng Fu, Tony Wu.

**Project administration:** Chia-Lin Wu.

**Supervision:** Chia-Lin Wu.

**Writing – original draft:** Chia-Lin Wu.

**Writing – review & editing:** Chia-Lin Wu.

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
