## [Editor Report · Decision Letter 0]

21 Apr 2023

Dear Dr Wu, 

Thank you for submitting your manuscript entitled "Independent insulin signaling modulators govern hot avoidance under different feeding states" for consideration as a Research Article by PLOS Biology.

Your manuscript has now been evaluated by the PLOS Biology editorial staff as well as by an academic editor with relevant expertise and I am writing to let you know that we would like to send your submission out for external peer review.

Once your full submission is complete, your paper will undergo a series of checks in preparation for peer review. After your manuscript has passed the checks it will be sent out for review. To provide the metadata for your submission, please Login to Editorial Manager (https://www.editorialmanager.com/pbiology) within two working days, i.e. by Apr 25 2023 11:59PM.

Kind regards,

Lucas

Lucas Smith, Ph.D.

Associate Editor

PLOS Biology

lsmith@plos.org

---

## [Decision Letter · Decision Letter 1]

8 Jun 2023

Dear Dr Wu,

Thank you for your patience while your manuscript "Independent insulin signaling modulators govern hot avoidance under different feeding states" was peer-reviewed at PLOS Biology. It has now been evaluated by the PLOS Biology editors, an Academic Editor with relevant expertise, and by several independent reviewers who find the study interesting and generally well done, but who have a number of suggestions to strengthen the study. In light of the reviews, which you will find at the end of this email, we would like to invite you to revise the work to thoroughly address the reviewers' reports.

Given the extent of revision needed, we cannot make a decision about publication until we have seen the revised manuscript and your response to the reviewers' comments. Your revised manuscript is likely to be sent for further evaluation by all or a subset of the reviewers.

**IMPORTANT - SUBMITTING YOUR REVISION**

*Re-submission Checklist*

*Published Peer Review*

*PLOS Data Policy*

*Blot and Gel Data Policy*

Sincerely,

Luke 

Lucas Smith, Ph.D.

Senior Editor

PLOS Biology

lsmith@plos.org

REVIEWS:

Reviewer #1: This manuscript uses behavioral assays and calcium imaging to understand how fed status affects hot avoidance behavior (HAB). The authors provide compelling evidence to support their findings. The authors first show hungry flies exhibit stronger HAB than fed flies. Then, they demonstrate different Dilps inhibit α'β' MBns in different fed statuses. Finally, they identify the neural circuits that control HAB from mALT projection neurons to α'β' MBns to MBONs. Experiments were carefully designed, and the manuscript was well-written. I enjoyed reading this manuscript. The following are two comments to help improve the manuscript.

1. Besides α'β', αβ and γ MBns are also necessary for HAB under satiety conditions (Fig 1E, F). How does fed status regulate αβ and γ MBns to regulate HAB in satiety conditions?

2. Considering that the inputs from mALT are similar in hungry and satiety conditions (Fig 5F), it is valuable to demonstrate whether the α'β' responses are also similar in both conditions in the absence of Dilp regulation. 

Reviewer #2: This study addresses the role of animal feeding status to thermosensation using Drosophila melanogaster as a model. The study shows convincingly that starved animal exhibit hot avoidance behavior compared to well fed animals. The study identified a group of mushroom body neurons that are responsible for the feeding-state dependent hot avoidance behavior. Moreover, neuros which mediate the heat input to the mushroom body neurons as well as output neurons mediating the hot avoidance behavior are characterized. The authors present evidence for a dual mode of regulation of these neurons through insulin-like receptor signaling. Insulin-like peptide (ILP) 2, secreted from the insulin-producing cell signal upon the fed state, while fat body-derived ILP6 signals the starved state. In sum, the study presents an elegant example of how insulin-like signaling, which is responsive to feeding state, interacts with animal behavior and provides a thorough characterization of the underlying neuronal mechanism. The experimental evidence presented in the study in extensive and the experiments are well controlled. Each main conclusion of the study is robustly supported by several lines of evidence. The findings are likely of broad scientific interest and therefore the manuscript is, in my opinion, suitable for Plos Biology.

I have one major concern that should be addressed prior to publication:

The authors propose a provocative model to explain their data of opposing functions of ILP2 and ILP6. The model implies that the fed state-induced signal by ILP2 is mediated through the PI3K/AKT pathway, the starvation-induced ILP6 signaling would activate the MAPK/ERK pathway. This is an elegant model, as it would explain how the Insulin-like receptor can mediate opposing signals in response to different ligands. However, to my knowledge, there is not known biochemical mechanism that would explain such ligand-specific signaling outputs. While the genetic manipulation of AKT vs ERK signaling as well as the pAKT and pERK signals upon fed and starved states, support the conclusion, the presented data does not sufficiently test the proposed model. Currently, it remains a valid possibility that ILP6 controls the mushroom body neurons indirectly.

Therefore, I request the following revisions:

1. Further experiments should be conducted to analyze the role of insulin-like signaling in the mushroom body neurons. Specifically, the authors should analyze, how pAKT and pERK signals change upon loss-of-function of InR, ILP2 and ILP6 in fed and starved animals. If necessary, the strength of conclusions should be adjusted according to the data. 

2. Analysis of AKT and ERK phosphorylation in fed and starved states is shown in supplementary data (Figure S7J and Figure S9N). This data, as well as the additional data from the requested experiments, should be displayed in a main figure.

Reviewer #3: Chiang et al. investigate the role of the mushroom body (MB) in temperature preference and food-seeking behaviors in Drosophila. They show that hungry flies exhibit stronger avoidance of hot stimuli compared to sated flies, while cold avoidance behavior is similar. The study identifies the neurons involved in conveying hot signals and reveals that MB activity is required for hot avoidance in both feeding states. Insulin signaling and specific Dilp signals play a role in regulating hot sensation and MB neuron activity. The study also identifies downstream circuits and signaling pathways involved in executing hot avoidance behavior. Overall, the authors suggest that distinct Dilp signals mediate MB neuron activity for proper hot avoidance behavior in different feeding states.

General comments:

In general, the paper is interesting and, in my opinion, could be appropriate for publication in PlosBiology provided that several issues are successfully addressed. The topic of how the perception of an external stimulus is changed by an internal state and the behavioral consequence of this altered perception is a very relevant topic. Clearly, the authors performed a remarkable quantity of experiments and manipulations. However, I would strongly suggest to simplify and make the story easier to follow. It would, for example, help in the results session to explain in more detail the reasoning behind the experiments. Moreover, a suggestion could be to partially rephrase the paper in terms of shifting of the setpoint and switch in valence of the hot stimulus rather than strong/weak hot avoidance behavior (Figure 1 C,E,G clearly show approach for 27 and 29 degrees). It is also very difficult to compare the relative differences between hungry and satiated flies across manipulations. The scale is bigger than the average effect (0-0.5-1 vs changes of maybe 0.1-0.2). Sometimes you have to believe the authors that starved flies are still doing more HAB than satiated ones. Finally, the role of dopaminergic neurons innervating the MB is somewhat ignored in the entire paper, which I find extremely surprising given the important role of these neurons in modulating (innate) behavior due to hunger and through MB output neurons (including in particular the b'1 compartment) and their response to temperature (see Tomchik, J Neurosci 2013, Bracker et al., Current Biology 2013, Siju et al., Current Biology 2020 etc.). Moreover, plenty of literature links dopamine, insulin and temperature in different processes in humans and rodents. Thus, this should be at least discussed in the text. 

Specific comments:

* Line 132: Fig S1 B-C are not addressed in the text 

* Line 147-148: not clear what the authors mean by temporally irradiated. In general, the protocol length and characteristics should be addressed more in detail. In the main text and methods it is stated that the total exp duration is 3 minutes and the flies were irradiated with light for the entire duration of the exp. However, in the Supplementary figure 1 E-F it is actually depicted as 3 min + 3 min (+ light) + 3 min.

* Line 207: Fig. S2H: would it not be a better control to puff cold water and show no response? After all, the difference in HAB happens because starved flies avoid hot temperatures more than WT but they behave similarly to WT for cold temperatures (Figure 1 B).

* Lines 217-223: maybe here the authors could clarify their reasoning in terms of expected consequences on behavior following depletion and over-expression of insulin receptors. HAB looks very different in Figure 1 C, D and Figure 2 A, B. The effects on behavior when the MB is manipulated vs InR are very different. 

* Lines 235-236: this directly relates to the core of the author's claim: without the modulation through insulin-like peptides there is no modulation in the MB of the hot stimulus response. Hungry flies should have the same level of activation as satiated flies. Therefore, the two conditions (satiety and hunger), now plotted separately Figure 2 I-J, should be plotted together as in Figure 1 H, I, J.

* Lines 278-279: Please be consistent with your statement in the introduction "levels of Dilp2 decrease in starved flies"

* Lines 281-282: the effects on HAB were present not only when alpha1beta1 was silenced but also when gamma and alphabeta were silenced in satiated flies (Figure 1 E F). Therefore, dilp2 phenotype in satiated flies should be checked in these lines, too. 

* Lines 285-286 and 295-296 are somewhat confusing. Please clarify what mediates hot responses by bringing information about the internal state of the animal. A more comprehensive explanation of the reasoning would be beneficial to the reader. 

* Lines 361-362: this is clearly not true as Figure 3 E shows that overexpression of Dilp2 results in a decrease of HAB, unless I have a misunderstanding…

* I suggest moving the Kir 2.1 manipulations to Supplementary to make panels easier to understand. 

* Finally, and although I try to not ask for new experiments, I would really like to see the 23 degrees and 25 degrees points in the plots as in Figure 1 B (perhaps the authors have those data already).

---

## [Decision Letter · Decision Letter 2]

25 Aug 2023

Dear Dr Wu,

Thank you for your patience while we considered your revised manuscript "Independent insulin signaling modulators govern hot avoidance under different feeding states" for publication as a Research Article at PLOS Biology. This revised version of your manuscript has been evaluated by the PLOS Biology editors, the Academic Editor and by reviewer 2. 

Both reviewer 2 and our Academic Editor are largely satisfied by the changes made in response to the previous reviewer comments, and both have suggested that we accept your study. However, before we formally accept your manuscript, we have a number of editorial requests which we need you to address in another revision that we think will not take very long. 

Please address the following editorial requests: 

1) The Academic Editor has one las suggestion which we think you should consider before publication. S/he says: "To future-proof your study, I suggest that you address the following editorial point, which does not require extra experiments but you may wish to consider it in the Discussion. Have you cross-checked the Ilp2 and Ilp6 manipulations with BOTH signalling pathways i.e not just p-Akt for Ilp2 and p-ERK for Ilp6? If not, it may be worth keeping open the possibility that Ilp2 may also increase p-ERK and Ilp6 also increase p-Akt?"

2) DATA REQUEST: Thank you for providing an S1_data file, which I assume has the underlying data related to your manuscript. As this was submitted as a .Rar file, I had trouble opening it. Can you please provide this as an excel file so I check that it meets our requirements?

>>Please do take a look at our data policy, and make sure the data provided meets our requirements, as this will be required before publication: http://journals.plos.org/plosbiology/s/data-availability. For more information, please also see this editorial: http://dx.doi.org/10.1371/journal.pbio.1001797

Note that we do not require all raw data. Rather, we ask that all individual quantitative observations that underlie the data summarized in the figures and results of your paper. Please ensure that you provide the individual numerical values that underlie the summary data displayed in your figures (including supplemental), as they are essential for readers to assess your analysis and to reproduce it.

>>Please also ensure that figure legends in your manuscript include information on where the underlying data can be found. For example, to each figure legend you can add the sentence "the data underlying this figure can be found in S1_data"

>>Please ensure your supplemental data file/s has a legend.

3) Per journal policy, any code that you have generated to support the conclusions of your manuscript should be made available without restrictions upon publication. If you used or generated code for this study, please ensure that the code is sufficiently well documented and reusable, and that your Data Statement in the Editorial Manager submission system accurately describes where your code can be found.

We expect to receive your revised manuscript within two weeks. 

*Published Peer Review History*

*Press*

Sincerely,

Luke

Lucas Smith, Ph.D.

Senior Editor,

lsmith@plos.org,

PLOS Biology

REVIEWS 

(note, we only sent the revision back to Reviewer 2, as the Academic Editor assessed the response to Reviewers 1 and 3)

Reviewer #2: The authors have addressed my comments in full and I have no further concerns. I wish to congratulate the authors for this interesting and rigorous study.

---

## [Editor Report · Decision Letter 3]

11 Sep 2023

Dear Dr Wu,

Thank you for the submission of your revised Research Article "Independent insulin signaling modulators govern hot avoidance under different feeding states" for publication in PLOS Biology and thank you for addressing our last editorial requests in this revision. On behalf of my colleagues and the Academic Editor, Alex P Gould, I am pleased to say that we can in principle accept your manuscript for publication, provided you address any remaining formatting and reporting issues. These will be detailed in an email you should receive within 2-3 business days from our colleagues in the journal operations team; no action is required from you until then. Please note that we will not be able to formally accept your manuscript and schedule it for publication until you have completed any requested changes.

**As you address any formatting and reporting requests, to come, please also address the following editorial requests, which I think was missed in my last letter: 

1) Thank you for updating your S1_data file, which meets our data sharing requirements. Please update the figure legends in your manuscript to reference this file. For example, to each figure legend (including supplemental) you can add the sentence "the data underlying this figure can be found in S1_data"

PRESS

Sincerely, 

Lucas Smith, Ph.D.

Senior Editor

PLOS Biology

lsmith@plos.org